# Randomized Sketches for Clustering:
# Fast and Optimal Kernel $k$-Means

**Rong Yin [1,2], Yong Liu [3,4],[*] Weiping Wang [1,2], Dan Meng [1,2]**

[1] Institute of Information Engineering, Chinese Academy of Sciences, Beijing, China
[2] School of Cyber Security, University of Chinese Academy of Sciences, Beijing, China
[3] Gaoling School of Artificial Intelligence, Renmin University of China, Beijing, China
[4] Beijing Key Laboratory of Big Data Management and Analysis Methods, Beijing, China
yinrong@iie.ac.cn, liuyonggsai@ruc.edu.cn, {wangweiping, mengdan}@iie.ac.cn

## Abstract

Kernel $k$-means is arguably one of the most common approaches to clustering. In this paper, we investigate the efficiency of kernel $k$-means combined with randomized sketches in terms of both statistical analysis and computational requirements. More precisely, we propose a unified randomized sketches framework to kernel $k$-means and investigate its excess risk bounds, obtaining the state-of-the-art risk bound with only a fraction of computations. Indeed, we prove that it suffices to choose the sketch dimension $\Omega(\sqrt{n})$ to obtain the same accuracy of exact kernel $k$-means with greatly reducing the computational costs, for sub-Gaussian sketches, the randomized orthogonal system (ROS) sketches, and Nyström kernel $k$-means, where $n$ is the number of samples. To the best of our knowledge, this is the first result of this kind for unsupervised learning. Finally, the numerical experiments on simulated data and real-world datasets validate our theoretical analysis.

## 1 Introduction

Kernel learning is an important field of machine learning Yin et al. (2020b,a, 2021, 2022). Kernel $k$-means is one of the fundamental approaches in unsupervised learning and has been widely used in numerous applications Zhang & Rudnicky (2002); Dhillon et al. (2004); Chitta et al. (2011); Li & Liu (2021), whose basic idea is to classify similar samples into the same cluster, and there is a large difference between samples in different clusters.

The statistical properties of kernel $k$-means have been studied for decades, but they may appear to be not sufficient. Consistency of the empirical minimizer of the clustering risk was shown in Abaya & Wise (1984); Pollard (1981, 1982). Rates of convergence and nonasymptotic performance bounds have been considered by Antos (2005); Antos et al. (2005); Bartlett et al. (1998); Linder (2000, 2002). The existing excess risk bounds are mostly dependent upon the dimension of the hypothesis space. For example, in Bartlett et al. (1998), the clustering risk upper bound is $\mathcal{O}(\sqrt{kd/n})$, where $n$ is the number of samples, $k$ is the number of clusters, and $d$ is the dimension of the hypothesis space. Note that the hypothesis space of kernel $k$-means is typically an infinite-dimensional Hilbert space and the upper bound become useless when $d$ is very large. Subsequently, some researchers deduced dimension-independent upper bounds for kernel $k$-means Koltchinskii (2006); Biau et al. (2008); Maurer & Pontil (2010); Canas et al. (2012); Levrard et al. (2015); Fefferman et al. (2016); Calandriello & Rosasco (2018); Liu (2021). However, the existing excess clustering risk bounds either have a slow convergence rate $\mathcal{O}(k/\sqrt{n})$ Biau et al. (2008); Calandriello & Rosasco (2018) or require pretty strong assumptions on the underlying distribution or large approximate dimensions $m$

---

[*]Corresponding author.

36th Conference on Neural Information Processing Systems (NeurIPS 2022).

to get the faster convergence rate. Specifically, in Calandriello & Rosasco (2018), if the approximate dimension reaches $\Omega(\sqrt{n})$, the clustering risk upper bound is $\mathcal{O}(k/\sqrt{n})$, which is proportional to $k$. Based on its method, Liu Yong Liu (2021) further improves the convergence rate to $\mathcal{O}(\sqrt{k/n})$, but the corresponding approximate dimension is increased to $\Omega(\sqrt{nk})$. Meanwhile, in order to reduce the approximate dimension to $\Omega(\sqrt{n})$ with the convergence rate unchanged, this paper Liu (2021) requires a stronger assumption of algebraically decreasing eigenvalues of the kernel matrix.

From the perspective of computational requirements, kernel $k$-means requires manipulating and storing an empirical kernel matrix, which is unfeasible for large-scale problems. Exploring approximate kernel $k$-means algorithms to scale to large-scale application scenarios has become a subject of recent works, see for example Nyström approximations Williams & Seeger (2001); Fowlkes et al. (2004); Pourkamali-Anaraki et al. (2018); Calandriello & Rosasco (2018); Wang et al. (2019); Liu (2021), randomized sketches Biau et al. (2008); Wang et al. (2019), random features Rahimi & Recht (2008); Chitta et al. (2012); Pham & Pagh (2013); Atarashi et al. (2019), incremental clustering Can (1993); Bradley et al. (2000), and reference therein. This paper focuses on the excess risk bound and computational requirements for kernel $k$-means. Although there are many studies on the approximate kernel $k$-means, these approximate works pay little attention to the excess risk of clusters with the exception of Biau et al. (2008); Calandriello & Rosasco (2018); Liu (2021). For example, the works in Wang et al. (2019) establish the $1 + \varepsilon$ relative-error bound for randomized sketches kernel $k$-means instead of excess risk bound. Therefore, in this paper, we mainly introduce the most related approximate kernel $k$-means with excess risk guarantees. In Biau et al. (2008), they employ the randomized sketches method to project the data in Hilbert space so as to approximate kernel $k$-means. However, the data in Hilbert space are implicit and infinite-dimensional, and its sketch matrix is dense and unstructured. In Calandriello & Rosasco (2018), the excess risk upper bound is $\mathcal{O}(k/\sqrt{n})$ when the approximate dimension reaches $\Omega(\sqrt{n})$. The upper bound of clustering risk in Biau et al. (2008) and Calandriello & Rosasco (2018) does not reach the optimal $\mathcal{O}(\sqrt{k/n})$ Bartlett et al. (1998). In Liu (2021), the approximate Nyström kernel $k$-means obtains the risk upper bound $\mathcal{O}(\sqrt{k/n})$ with the approximate dimension $\Omega(\sqrt{nk})$. Although this paper Liu (2021) further reduces the approximate dimension to $\Omega(\sqrt{n})$ by introducing a stronger assumption, this is not universal. In addition, the computational requirements in Biau et al. (2008); Calandriello & Rosasco (2018); Liu (2021) are still high.

Motivated by these issues, in this paper, we focus on improving the statistical analysis and computational approximations of kernel $k$-means. We propose a randomized sketches framework to kernel $k$-means and construct three novel and specific examples: sub-Gaussian sketches, the randomized orthogonal system (ROS) sketches, and Nyström kernel $k$-means. Theoretical analysis shows that the proposed three randomized sketches methods obtain the optimal excess clustering risk upper bound $\mathcal{O}(\sqrt{k/n})$ with the sketch dimension (i.e. approximate dimension) of $\Omega(\sqrt{n})$ (see Theorem 2). To the best of our knowledge, this is the first optimal excess risk bound with the least approximate dimension and no strong assumptions for general approximate kernel $k$-means. From a computational point of view, the proposed methods lead to massive improvements reducing the time complexity from $\mathcal{O}(n^2 kt)$ to at least $\mathcal{O}(n\sqrt{n} + n\sqrt{nk}t)$ and the memory complexity from $\mathcal{O}(n^2)$ to $\mathcal{O}(n\sqrt{n})$, where $t$ is the number of iterations. Moreover, we further derive the similarity bound of approximate solutions in the general case, which can be effectively calculated by $k$-means++ (see Theorem 3). Experimental results verify and illustrate our theoretical analysis.

The rest of the paper is organized as follows. Section 2 is the background of kernel $k$-means. Section 3 describes the proposed randomized sketches kernel $k$-means framework and provides three novel examples. In section 4, we mainly show excess risk bounds of the proposed randomized sketches kernel $k$-means and the further theoretical analysis in the general case of $k$-means++. Sections 5 and 6 are the experiments and conclusions.

## 2 Background

### 2.1 Notation

Given a sampling distribution $\mu$ on an arbitrary input space $\mathcal{X}$ and $n$ samples $\mathcal{S} = \{\mathbf{x}_i\}_{i=1}^{n} \in \mathcal{X}$ drawn i.i.d. from $\mu$, we denote with $\mu_n(\mathcal{S}) = \frac{1}{n} \sum_{i=1}^{n} \mathbb{I}\{\mathbf{x}_i \in \mathcal{X}\}$ the empirical distribution, where $\mathbb{I}(\cdot)$ is the indicator function. In this paper, we use the feature map $\varphi(\cdot) : \mathcal{X} \to \mathcal{H}$ to map $\mathcal{X}$

into a Reproducing Kernel Hilbert Space (RKHS) $\mathcal{H}$ Schölkopf et al. (2002); Scholkopf & Smola (2018), and assume that $\mathcal{H}$ is separable, such that for any $\mathbf{x} \in \mathcal{X}$, we have $\Phi_{\mathbf{x}} = \varphi(\mathbf{x})$. Let $\kappa : \mathcal{X} \times \mathcal{X} \to \mathbb{R}$ be a mercer kernel. We denote the inner product of $\mathcal{H}$ by $\langle \cdot, \cdot \rangle$, the associated norm by $\| \cdot \|$, the Cartesian product of $\mathcal{H}$ by $\mathcal{H}^k = \otimes_{i=1}^k \mathcal{H}$, and with $\mathbf{K}$ the kernel matrix, where $\mathbf{K}_{ij} = \kappa(\mathbf{x}_i, \mathbf{x}_j) = \langle \Phi_i, \Phi_j \rangle = \Phi_i^T \Phi_j$. This paper assumes that $\|\Phi_{\mathbf{x}}\| \le 1$ for any $\mathbf{x} \in \mathcal{X}$.

## 2.2 Kernel $k$-Means

Let $\mathbf{C} = [\mathbf{c}_1, \dots, \mathbf{c}_k]$ be a collection of $k$ centroids from $\mathcal{H}$. We divide the given dataset into $k$ disjoint clusters, each characterized by its centroid $\mathbf{c}_j$. The Voronoi cell associated with a centroid $\mathbf{c}_j$ is defined as Calandriello & Rosasco (2018)

$$\mathcal{C}_j := \{i : j = \arg \min_{s=[k]} \|\Phi_i - \mathbf{c}_s\|^2\}, \tag{1}$$

where $[k] = 1, 2, \dots, k$. That is, the point $\Phi_i$ belongs to the $j$-th cluster if $\mathbf{c}_j$ is its closest centroid. Now we formalize the criterion used to measure the clustering quality. The empirical squared norm criterion is defined as

$$W(\mathbf{C}, \mu_n) := \frac{1}{n} \sum_{i=1}^n \min_{j=[k]} \|\Phi_i - \mathbf{c}_j\|^2, \tag{2}$$

and the expected squared norm criterion is defined as

$$W(\mathbf{C}, \mu) := \mathbb{E}_{\Phi \sim \mu}[\min_{j=[k]} \|\Phi - \mathbf{c}_j\|^2]. \tag{3}$$

The empirical risk minimizer (ERM) is defined as

$$\mathbf{C}_n := \arg \min_{\mathbf{C} \in \mathcal{H}^k} W(\mathbf{C}, \mu_n). \tag{4}$$

The sub-script $n$ in $\mathbf{C}_n$ indicates that it minimizes $W(\mathbf{C}, \mu_n)$ for $n$ samples in $\mathcal{S}$.

In this paper, we bound the excess clustering risk $\mathcal{E}(\mathbf{C}_n)$ of the empirical risk minimizer Calandriello & Rosasco (2018):

$$\mathcal{E}(\mathbf{C}_n) := \mathbb{E}_{\mathcal{S} \sim \mu}[W(\mathbf{C}_n, \mu)] - W^*(\mu), \tag{5}$$

where $W^*(\mu) := \inf_{\mathbf{C} \in \mathcal{H}^k} W(\mathbf{C}, \mu)$ is the optimal clustering risk. In the following, we will ignore the subscript $\mathcal{S} \sim \mu$ if the input dataset $\mathcal{S}$ is clear.

From a computational perspective, one cannot compute $\mathbf{C}_n$ directly, since the points $\Phi_i$ in $\mathcal{H}$ cannot be explicitly represented. However, due to the properties of the squared norm criterion and the kernel trick, one can reformulate the objective $W(\cdot, \mu_n)$ of kernel $k$-means.

**Proposition 1 (Proposition 2 of Calandriello & Rosasco (2018)).** *Let $\mathbf{K}_{nn} \in \mathbb{R}^{n \times n}$ be the empirical kernel matrix, and $\mathbf{k}_i$ its $i$-th columns. Then*

$$
\begin{aligned}
\min_{\mathbf{C} \in \mathcal{H}} W(\mathbf{C}, \mu_n) &= \frac{1}{n} \min_{\nu} \sum_{j=1}^k \sum_{i \in \mathcal{C}_j} \left\| \Phi_i - \frac{1}{|\mathcal{C}_j|} \sum_{s \in \mathcal{C}_j} \Phi_s \right\|^2 \\
&= \frac{1}{n} \min_{\nu} \sum_{j=1}^k \sum_{i \in \mathcal{C}_j} \left\| \mathbf{k}_i - \frac{1}{|\mathcal{C}_j|} \sum_{s \in \mathcal{C}_j} \mathbf{k}_s \right\|^2.
\end{aligned}
\tag{6}
$$

This approach constructs an $n$-dimensional embedding $\mathbf{k}_i$ for each point $i$, namely the $i$-th columns of the kernel matrix $\mathbf{K}_{nn}$, which can be explicitly computed, and perfectly preserves $W(\cdot, \mu_n)$ and its minimizer $\mathbf{C}_n$. However, it requires $\mathcal{O}(n^2)$ time and space to construct and store the kernel matrix $\mathbf{K}$, which is not scalable to large-scale scenarios.

## 2.3 The Existing Excess Risk Bounds of Kernel $k$-Means

Here we provide the existing upper bound and lower bound of kernel $k$-means.

According to Bartlett et al. (1998), we know that there exists a collection of centroids $\mathbf{C}_l \in \mathcal{H}^k$ and $\|\Phi_{\mathbf{x}}\| \leq 1$ for any $\mathbf{x} \in \mathcal{X}$, such that

$$\mathbb{E}[W(\mathbf{C}_l, \mu)] - W^*(\mu) = \Omega\left(\sqrt{\frac{k^{1-4/d}}{n}}\right), \tag{7}$$

where $d$ is the dimension of $\Phi_{\mathbf{x}}$. In general, $d$ is very large or even infinite. Therefore, the lower bound of kernel $k$-means is $\Omega\left(\sqrt{\frac{k}{n}}\right)$. The following is the upper bound of kernel $k$-means.

**Theorem 1** (**Theorem 1 in Liu (2021)**). *If $\|\Phi_{\mathbf{x}}\| \leq 1$ for any $\mathbf{x} \in \mathcal{X}$, then for any $\delta \in (0, 1)$, with probability at least $1 - \delta$, we have,*

$$\mathcal{E}(\mathbf{C}_n) = \mathbb{E}[W(\mathbf{C}_n, \mu)] - W^*(\mu)$$
$$= \mathcal{O}\left(\sqrt{\frac{k}{n}} \log^2(\sqrt{n})\right) = \tilde{\mathcal{O}}\left(\sqrt{\frac{k}{n}}\right). \tag{8}$$

Note that, $p = \mathcal{O}(u)$ means that there exists an constant $c$ such that $p \leq cu$. $\tilde{\mathcal{O}}(\cdot)$ means to hide the logarithmic terms. This upper bound matches the theoretical lower bound $\Omega\left(\sqrt{\frac{k}{n}}\right)$, and therefore shows that the ERM $\mathbf{C}_n$ achieve an excess risk (nearly) optimal in $n$.

## 3 The Proposed Algorithms

Kernel $k$-means is one of the most popular clustering methods Yin et al. (2020c). However, it is non-scalable to large scenarios due to computing the exact embedding $\mathbf{k}_i$. To reduce the computational requirements, we propose novel approximate embeddings by using randomized sketches. In this section, we propose a unified randomized sketches kernel $k$-means. In addition, three specific examples of randomized sketches algorithms and the corresponding complexity analysis are provided.

### 3.1 Framework of Randomized Sketches Kernel $k$-Means

We consider an approximation based on reducing the original column $\mathbf{k}_i \in \mathbb{R}^n$ to an $m$-dimensional subspace of $\mathbb{R}^n$, where $m \ll n$ is the sketch dimension. More precisely, the proposed approximation is defined via a sketch matrix $\mathbf{R} \in \mathbb{R}^{m \times n}$, such that the $m$-dimensional subspace is generated by the span of $\mathbf{R}$. Therefore, the proposed randomized sketches method can be described as:

$$\tilde{\mathbf{K}} = \mathbf{R}\mathbf{K} = \mathbf{S}\mathbf{Q}\mathbf{K} \in \mathbb{R}^{m \times n}, \tag{9}$$

where $\mathbf{K} = \mathbf{K}_{nn}$ and $\mathbf{Q} \in \mathbb{R}^{m \times n}$ is a sampling matrix. The rows of $\mathbf{Q}$ are composed of $m$ rows sampled uniformly from the $n \times n$ identity matrix without replacement. The matrix $\mathbf{S} \in \mathbb{R}^{m \times m}$ is constructed in three ways, which will be introduced in detail in the following section.

Then the unified randomized sketches kernel $k$-means can be written as (similarly to Proposition 1):

$$\bar{\mathbf{C}}_{n,m} = \arg\min_{\bar{\mathbf{C}} \in \mathbb{R}^{m \times k}} \frac{1}{n} \sum_{i=1}^{n} \min_{j=1,\dots,k} \left\| \tilde{\mathbf{k}}_i - \bar{\mathbf{c}}_j \right\|^2$$
$$= \frac{1}{n} \min_{\nu} \sum_{j=1}^{k} \sum_{i \in \mathcal{C}_j} \left\| \tilde{\mathbf{k}}_i - \frac{1}{|\mathcal{C}_j|} \sum_{s \in \mathcal{C}_j} \tilde{\mathbf{k}}_s \right\|^2, \tag{10}$$

where $\tilde{\mathbf{k}}_i$ is the column of the approximate kernel matrix $\tilde{\mathbf{K}}$ in Eq.(9) and $\bar{\mathbf{C}}_{n,m} = [\bar{\mathbf{c}}_1, \dots, \bar{\mathbf{c}}_k]$ is the empirical clustering centers associated with the $m$-dimensional $\tilde{\mathbf{k}}_1, \dots, \tilde{\mathbf{k}}_n$. Each $\bar{\mathbf{c}}_j$ is the mean of those $\tilde{\mathbf{k}}_i$'s in the Voronoi cell $\tilde{\mathcal{C}}_j$.

Define the clustering centers by

$$\tilde{\mathbf{c}}_j = \frac{\sum_{i=1}^{n} \mathbf{k}_i \mathbb{I}_{\{\tilde{\mathbf{k}}_i \in \tilde{\mathcal{C}}_j\}}}{\sum_{i=1}^{n} \mathbb{I}_{\{\tilde{\mathbf{k}}_i \in \tilde{\mathcal{C}}_j\}}}, \quad j = 1, \dots, k, \tag{11}$$

where $\tilde{\mathbf{C}}_{n,m} = [\tilde{\mathbf{c}}_1, \ldots, \tilde{\mathbf{c}}_j, \ldots, \tilde{\mathbf{c}}_k]$ and $\mathbb{I}_{\{\cdot\}}$ is the indicator function. $\mathbb{I}_{\{\tilde{\mathbf{k}}_i \in \tilde{\mathcal{C}}_j\}} = 1$ if $\tilde{\mathbf{k}}_i \in \tilde{\mathcal{C}}_j$ and $\mathbb{I}_{\{\tilde{\mathbf{k}}_i \in \tilde{\mathcal{C}}_j\}} = 0$ otherwise.

The $n$-dimensional embeddings $\mathbf{k}_i$ are replaced with the lower $m$-dimensional enbeddings $\tilde{\mathbf{k}}_i$. We can perform any $k$-means algorithms over $\{\tilde{\mathbf{k}}_1, \ldots, \tilde{\mathbf{k}}_n\}$ and compute the clustering centers in Eq.(11).

---

**Algorithm 1** Unified Randomized Sketches Kernel $k$-Means

**Input**: dataset $\mathcal{S} = \{\mathbf{x}_i\}_{i=1}^n$, number of clusters $k$, kernel parameter, and sketch dimension $m$.
**Output**: centroids $\tilde{\mathbf{C}}_{n,m}$.

1: Sample $m$ data points from $\mathcal{S}$ according to the sampling matrix $\mathbf{Q}$ in Eq.(9).
2: Compute the approximate kernel matrix $\hat{\mathbf{K}} \in \mathbb{R}^{m \times n}$ between the $m$ sampling data points and the all data points in $\mathcal{S}$.
3: Construct the matrix $\mathbf{S} \in \mathbb{R}^{m \times m}$ (See Section 3.2 for specific construction methods).
4: Compute $\mathbf{S}\hat{\mathbf{K}}$, namely $\mathbf{S}\hat{\mathbf{K}} = \mathbf{S}(\mathbf{Q}\mathbf{K}) = \tilde{\mathbf{K}}$ is Eq.(9).
5: Perform $k$-means algorithm over the columns of $\tilde{\mathbf{K}}$.
6: Compute centroids $\tilde{\mathbf{C}}_{n,m}$ in Eq.(11).

---

The detail of the proposed randomized sketches kernel $k$-means is shown in Algorithm 1. The proposed algorithm is mainly divided into two parts. The first part is from step 1 to step 4, which is mainly to construct the sketch matrix $\mathbf{R} = \mathbf{S}\mathbf{Q}$ and obtain the variant kernel matrix $\tilde{\mathbf{K}} = \mathbf{S}\mathbf{Q}\mathbf{K}$. The second part is from step 5 to step 6, mainly performing $k$-means over the columns of $\tilde{\mathbf{K}}$ and obtaining centroids. In step 1, one samples $m$ data points from $\mathcal{S}$ according to the sampling matrix $\mathbf{Q}$. Then, computing the variant kernel matrix $\hat{\mathbf{K}} \in \mathbb{R}^{m \times n}$ by $m$ sampling data points and all $n$ data points. From a mathematical point of view, this step can be expressed as $\hat{\mathbf{K}} = \mathbf{Q}\mathbf{K}$. In step 3, we construct a matrix $\mathbf{S}$, whose specific expression will be given in Section 3.2. This paper provides three different examples of $\mathbf{S}$, which brings different effects in the approximate kernel $k$-mean algorithms. In step 5, take the columns $\tilde{\mathbf{k}}_i$ of $\tilde{\mathbf{K}} = \mathbf{S}\hat{\mathbf{K}}$ generated in step 4 as the processing objects and execute $k$-means algorithm on them. Finally, compute the centroids $\tilde{\mathbf{C}}_{n,m}$ in Eq.(11).

### 3.2 Examples of Randomized Sketches Kernel $k$-Means

Here, we introduce three examples of randomized sketches kernel $k$-means, which are constructed by three different matrices $\mathbf{S}$ in Eq.(9). In addition, the detailed complexity analysis of the corresponding three approximate kernel $k$-means is provided.

**Example 1: Sub-Gaussian Sketches Kernel $k$-Means** The first example of approximate kernel $k$-means is called sub-Gaussian sketches kernel $k$-means, whose matrix $\mathbf{S} \in \mathbb{R}^{m \times m}$ in Eq.(9) is described by a hash function. Let $\sigma$ be a hash function and $\sigma(i) \in \{+1, -1\}$ is 2-wise independent hash function. The entries $\mathbf{S}_{i,j} = \sigma(i)/\sqrt{m}$ with a probability of $\frac{1}{\sqrt{n}}$ and $\mathbf{S}_{i,j} = 0$ with a probability of $1 - \frac{1}{\sqrt{n}}$.

Complexity analysis: In the terms of time, we first sample the data by $\mathbf{Q}$, then generate the variant kernel matrix $\hat{\mathbf{K}}$. Therefore, the time cost of computing $\mathbf{S}\hat{\mathbf{K}}$ should be $\mathcal{O}(nm^2)$. However, due to the sparsity of the sub-Gaussian matrix $\mathbf{S}$, we only need to compute the non-zero elements instead of the total elements, which can further reduce the computational requirements of $\mathbf{S}\hat{\mathbf{K}}$ from $\mathcal{O}(nm^2)$ to $\mathcal{O}(\sqrt{n}m^2)$. In the iteration operation of performing $k$-means algorithm over the columns $\tilde{\mathbf{k}}_i$ of $\tilde{\mathbf{K}}$, one needs $\mathcal{O}(nmkt)$ time. Combining the above, the total time cost of sub-Gaussian sketches kernel $k$-means is $\mathcal{O}(\sqrt{n}m^2 + nmkt)$. In terms of space, due to the operation of sampling, the key of the space cost is changed to $\hat{\mathbf{K}}$ and $\tilde{\mathbf{K}}$ instead of $\mathbf{K}$. Therefore, the space complexity of the proposed sub-Gaussian sketches kernel $k$-means is $\mathcal{O}(nm)$.

**Example 2: ROS Sketches Kernel $k$-Means** The second example of the random sketches kernel $k$-means is based on the randomized orthogonal system (ROS) sketches. The corresponding matrix

$\mathbf{S} \in \mathbb{R}^{m \times m}$ in Eq.(9) can be defined as below:

$$\mathbf{S} = \mathbf{DA}, \tag{12}$$

where $\mathbf{D} \in \mathbb{R}^{m \times m}$ is a random diagonal matrix whose entries are i.i.d. Rademacher variables. $\mathbf{A} \in \mathbb{R}^{m \times m}$ is an orthogonal matrix with uniformly bounded entries, for example the Hadamard matrix Wallis (1976) and the discrete Fourier transform matrix. We use the Hadamard matrix in this paper. The Hadamard matrix is defined recursively as: $\mathbf{A}_m = \begin{bmatrix} \mathbf{A}_{m/2} & \mathbf{A}_{m/2} \\ \mathbf{A}_{m/2} & -\mathbf{A}_{m/2} \end{bmatrix}$ with $\mathbf{A}_2 = \begin{bmatrix} 1 & 1 \\ 1 & -1 \end{bmatrix}$, and $\mathbf{A} = \frac{1}{\sqrt{m}} \mathbf{A}_m$.

Due to the constructed property of the Hadamard matrix $\mathbf{A}$, we can use FFT (Fast Fourier Transform algorithm) to compute the matrix-vector product, such as $\mathbf{Au}$ for any $\mathbf{u} \in \mathbb{R}^m$, whose time complexity is $\mathcal{O}(m \log m)$ instead of $\mathcal{O}(m^2)$. Therefore, in step 4 of Algorithm 1, the computation of $\mathbf{S}\hat{\mathbf{K}}$ can be realized by the fast FFT, which is another way to further reduce the time cost, in addition to the sparsity mentioned above.

Complexity analysis: In terms of time cost, due to the use of the Hadamard matrix, we can compute $\mathbf{S}\hat{\mathbf{K}}$ by FFT, whose time cost is $\mathcal{O}(nm \log m)$. In the iteration operation of $k$-means algorithm over the columns $\tilde{\mathbf{k}}_i$, the time cost is $\mathcal{O}(nmkt)$. Therefore, the total time cost of ROS sketches kernel $k$-means is $\mathcal{O}(nm \log m + nmkt)$. In terms of space, the key to the space cost is to store the matrices $\hat{\mathbf{K}}$ and $\tilde{\mathbf{K}}$, whose space requirements is $\mathcal{O}(nm)$. Therefore, the space complexity of the proposed ROS sketches kernel $k$-means is $\mathcal{O}(nm)$.

**Example 3: Nyström Kernel $k$-Means**  The third example of the approximate kernel $k$-means is Nyström kernel $k$-means, whose matrix $\mathbf{S} \in \mathbb{R}^{m \times m}$ in Eq.(9) can be defined as: $\mathbf{S} = \mathbf{I}$, where $\mathbf{I}$ is an identify matrix. That is, we only use the sample matrix $\mathbf{Q}$ in Eq.(9). Therefore, the proposed Nyström kernel $k$-means can be converted into:

$$\begin{aligned} \tilde{\mathbf{C}}_{n,m} &= \arg\min_{\tilde{\mathbf{C}} \in \mathbb{R}^{m \times k}} \frac{1}{n} \sum_{i=1}^{n} \min_{j=1,\dots,k} \left\| \tilde{\mathbf{k}}_i - \tilde{\mathbf{c}}_j \right\|^2 \\ &= \arg\min_{\tilde{\mathbf{C}} \in \mathbb{R}^{m \times k}} \frac{1}{n} \sum_{i=1}^{n} \min_{j=1,\dots,k} \left\| \tilde{\Phi}_i - \tilde{\mathbf{c}}_j \right\|^2, \end{aligned} \tag{13}$$

where $\tilde{\Phi}_i = \mathbf{\Phi}_m^T \Phi_i$, $\tilde{\mathbf{c}}_j = \mathbf{\Phi}_m^T \mathbf{c}_j$, $\mathbf{\Phi}_m = [\Phi_{\pi(1)}, \dots, \Phi_{\pi(m)}]$, $\pi(i) \in [1, n]$, and the dictionary (i.e., subset) $\{\Phi_{\pi(i)}\}_{i=1}^{m}$ is $m$ points $\Phi_j$ sampled from $\{\Phi_j\}_{j=1}^{n}$ through the sampling matrix $\mathbf{Q}$.

Note that, the proposed Nyström kernel $k$-means can also be understood as a variant ROS, based on the identity matrix as an orthonormal matrix and not using the Rademacher randomization.

Complexity analysis: In terms of time cost, the matrix $\mathbf{S}$ is a scaled identity matrix so that the computation of step 3 and step 4 in Algorithm 1 is not needed. Therefore, the time complexity of Nyström kernel $k$-means is decided by the iteration operation of $k$-means algorithm over the columns $\tilde{\mathbf{k}}_i$, which is $\mathcal{O}(nmkt)$. In terms of space, the key is the matrix $\tilde{\mathbf{K}}$. Therefore, the space complexity of the proposed Nyström kernel $k$-means is $\mathcal{O}(nm)$.

In algorithm, the function of $\mathbf{Q}$ is to reduce the scale of data. The function of $\mathbf{S}$ is to fuse data features. In complexity, the proposed randomized sketches can reduce the time and space complexity. We sample data points according to $\mathbf{Q}$, then generate the variant kernel matrix, instead of generating and processing the kernel matrix directly, which can greatly reduce the time and space complexity. In addition, our matrices $\mathbf{S}$ are structured (in ROS) or sparse (in sub-Gaussian and Nyström), which can speed up kernel $k$-means by FFT or sparsity. In theoretical analysis, we obtain the optimal excess risk bound with a small sketch dimension based on the proposed randomized sketches, which can further reduce the time and space complexity. Overall, the proposed randomized sketches can greatly reduce the time and space complexity with the optimal excess risk bound.

## 4 Theoretical Analysis

In this section, we exploit the excess risk bound of the proposed randomized sketches kernel $k$-means. Theoretical analysis shows that we can improve the computational requirements of kernel $k$-means using sub-Gaussian, ROS, and Nyström, while maintaining optimal generalization guarantees.

**Theorem 2.** *If $\|\Phi_{\mathbf{x}}\| \leq 1$ for any $\mathbf{x} \in \mathcal{X}$, $\varepsilon \in (0,1)$, $\delta \in (0,1)$, and, in either one of the three cases of sub-Gaussian, ROS, and Nyström, the sketch dimension is $m = \Omega\left(\frac{4\log n - 2\log\delta}{\varepsilon - \log(1+\varepsilon)}\right)$, then, with probability at least $1 - \delta$, we have*

$$\mathbb{E}[W(\tilde{\mathbf{C}}_{n,m},\mu)] - W^*(\mu) = \tilde{\mathcal{O}}\left(\sqrt{\frac{k}{n}}\right) + \mathcal{O}\left(\frac{\varepsilon}{1-\varepsilon}\right). \tag{14}$$

**Remark 1.** *From a statistical point of view, let $\varepsilon = 1/\sqrt{n}$, Theorem 2 shows that when the sketch dimension is $m = \Omega(\sqrt{n})$, the proposed randomized sketches (sub-Gaussian, ROS, and Nyström) kernel $k$-means achieve the same excess risk bound $\tilde{\mathcal{O}}\left(\sqrt{k/n}\right)$ as the exact kernel $k$-means.*

**Remark 2.** *From a computational point of view, we can construct the $\sqrt{n}$-dimension randomized sketches simply, which can greatly reduces the total required space from $\mathcal{O}(n^2)$ to $\mathcal{O}(n\sqrt{n})$ and the total required time from $\mathcal{O}(n^2kt)$ to $\mathcal{O}(n\sqrt{n} + n\sqrt{n}kt)$ at least, with the optimal excess risk bound.*

**Remark 3.** *In Calandriello & Rosasco (2018), when the approximate dimension $m$ is $\Omega(\sqrt{n})$, the excess risk bound can reach $\tilde{\mathcal{O}}(k/\sqrt{n})$, which is linearly dependent on $k$ and fail to reach the optimal bound. The corresponding space complexity and time complexity are $\mathcal{O}(n\sqrt{n})$ and $\mathcal{O}(nkt\sqrt{n} + n^2)$, respectively. Compared to it, our proposed methods obtain the better excess risk bound and reduce the time complexity from $\mathcal{O}(nkt\sqrt{n} + n^2)$ to $\mathcal{O}(nkt\sqrt{n} + n\sqrt{n})$ at least. Subsequently, Liu Yong Liu (2021) further improves the excess risk bound of the method in Calandriello & Rosasco (2018) to $\tilde{\mathcal{O}}\left(\sqrt{k/n}\right)$, but the corresponding approximate dimension $m$ is increased to $\Omega(\sqrt{nk})$. Meanwhile, its space complexity and time complexity increase to $\mathcal{O}\left(n\sqrt{nk}\right)$ and $\mathcal{O}\left(nkt\sqrt{nk} + n^2k\right)$. Compared to it, the proposed methods reduce the time complexity from $\mathcal{O}\left(nkt\sqrt{nk} + n^2k\right)$ to $\mathcal{O}\left(nkt\sqrt{n} + n\sqrt{n}\right)$ at least and reduce the space complexity from $\mathcal{O}\left(n\sqrt{nk}\right)$ to $\mathcal{O}\left(n\sqrt{n}\right)$ while maintaining the optimal excess risk bound $\tilde{\mathcal{O}}\left(\sqrt{k/n}\right)$ and the smaller $m = \Omega(\sqrt{n})$. To the best of our knowledge, the proposed methods are the first time that they are always possible to maintain the optimal excess risk bound $\tilde{\mathcal{O}}\left(\sqrt{k/n}\right)$ in unsupervised non-parametric problem with smaller $m = \Omega\left(\sqrt{n}\right)$, while greatly reducing the time and space requirements. In Table 1, we show the detail space complexity, time complexity, excess risk bounds, and $m$ of the approximate kernel $k$-means.*

### 4.1 Further Results: $k$-Means++

We adopt the improved kernel $k$-means++ sampling Lattanzi & Sohler (2019), which has a local search strategy, for the proposed randomized sketches kernel $k$-mean. Here is its theoretical analysis.

**Lemma 1** (**Lattanzi & Sohler (2019)**). *If $C_n^+$ is obtained by the improved $k$-means++ algorithm with a local search strategy Lattanzi & Sohler (2019), then $\mathbb{E}_{\mathcal{J}}\left[W(\mathbf{C}_n^+, \mu_n)\right] \leq \varpi \cdot W(\mathbf{C}_n, \mu_n)$, where $\varpi$ is a constant and $\mathcal{J}$ is the randomness derived from the $k$-means++ initialization.*

Note that, this is a multiplicative error bound on the empirical risk.

**Theorem 3.** *Let $C_{n,m}^+$ be obtained by the improved $k$-means++ algorithm with a local search strategy Lattanzi & Sohler (2019). If $\|\Phi_{\mathbf{x}}\| \leq 1$ for any $\mathbf{x} \in \mathcal{X}$, $\varepsilon \in (0,1)$, $\delta \in (0,1)$, and, in either one of the three cases of sub-Gaussian, ROS, and Nyström, the sketch dimension is $m = \Omega\left(\frac{4\log n - 2\log\delta}{\varepsilon - \log(1+\varepsilon)}\right)$, then, with probability at least $1 - \delta$, we have*

$$\mathbb{E}_{\mathcal{S}}\left[\mathbb{E}_{\mathcal{J}}\left[W(\mathbf{C}_{n,m}^+, \mu)\right]\right] = \tilde{\mathcal{O}}\left(\sqrt{\frac{k}{n}} + W^*(\mu)\right) + \mathcal{O}\left(\frac{\varepsilon}{1-\varepsilon}\right), \tag{15}$$

*where $\mathcal{J}$ is the randomness derived from the $k$-means++ initialization.*

Table 1: Comparison of the approximate kernel $k$-means. The second and third columns represent the space and time complexity. The fourth and fifth columns represent the excess risk bounds and $m$.

| Approach | Space | Time | Bound | $m$ |
|---|---|---|---|---|
| Kernel $k$-Means | $\mathcal{O}\left(n^2\right)$ | $\mathcal{O}\left(n^2 kt\right)$ | $\tilde{\mathcal{O}}\left(\sqrt{\frac{k}{n}}\right)$ | / |
| NytrömCalandriello & Rosasco (2018) | $\mathcal{O}\left(n\sqrt{n}\right)$ | $\mathcal{O}\left(nkt\sqrt{n}+n^2\right)$ | $\tilde{\mathcal{O}}\left(\frac{k}{\sqrt{n}}\right)$ | $\sqrt{n}$ |
| NytrömLiu (2021) | $\mathcal{O}\left(n\sqrt{nk}\right)$ | $\mathcal{O}\left(nkt\sqrt{nk}+n^2 k\right)$ | $\tilde{\mathcal{O}}\left(\sqrt{\frac{k}{n}}\right)$ | $\sqrt{nk}$ |
| Sub-Gaussian Sketches (**This Paper**) | $\mathcal{O}\left(n\sqrt{n}\right)$ | $\mathcal{O}\left(nkt\sqrt{n}+n\sqrt{n}\right)$ | $\tilde{\mathcal{O}}\left(\sqrt{\frac{k}{n}}\right)$ | $\sqrt{n}$ |
| ROS Sketches (**This Paper**) | $\mathcal{O}(n\sqrt{n})$ | $\mathcal{O}(nkt\sqrt{n}+n\sqrt{n})$ | $\tilde{\mathcal{O}}\left(\sqrt{\frac{k}{n}}\right)$ | $\sqrt{n}$ |
| Nyström (**This Paper**) | $\mathcal{O}\left(n\sqrt{n}\right)$ | $\mathcal{O}\left(nkt\sqrt{n}\right)$ | $\tilde{\mathcal{O}}\left(\sqrt{\frac{k}{n}}\right)$ | $\sqrt{n}$ |

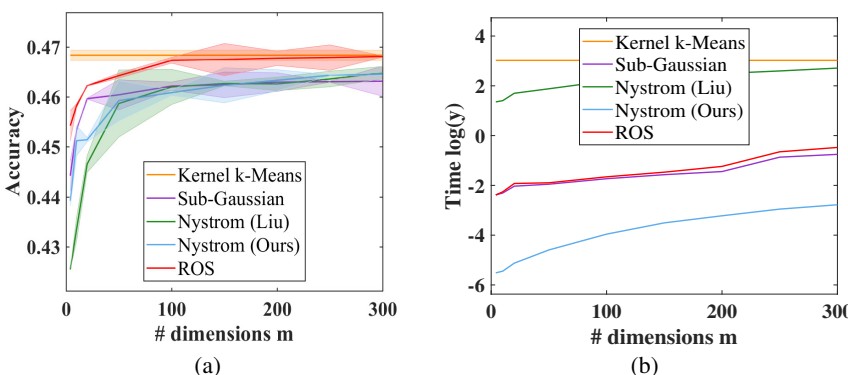

(a)  (b)

Figure 1: Test accuracy and training time (in seconds) with different dimensions $m$ of kernel $k$-means, Sub-Gaussian, ROS, Nystrom (Ours), and Nystrom (Liu) on simulated data.

Theorem 3 shows that, if the optimal clustering risk $W^*(\mu)$ is smaller than $\tilde{\mathcal{O}}(\sqrt{k/n})$, the risk of $W(\mathbf{C}_{n,m}^+,\mu)$ can reach $\tilde{\mathcal{O}}(\sqrt{k/n})$. Note that $\varepsilon$ is small, i.e. $\varepsilon = 1/\sqrt{n}$.

## 5   Experiments

In this section, we evaluate experimentally our theoretical analysis on both simulated data and real-world data for the proposed methods. The server is 32 cores (2.40GHz) and 32 GB of RAM. The compared methods are the exact kernel $k$-means, Gaussian Biau et al. (2008), Nyström (Liu) Liu (2021), ROS, Sub-Gaussian, and Nyström (Ours). For the sake of distinguishing, Nyström (Liu) is Nyström Liu (2021) in this paper. Each experiment is repeated 5 times.

### 5.1   Numerical Experiments on Simulated Data

We conduct the experiments to validate our theoretical analysis of the proposed randomized sketches kernel $k$-means on simulated data. Now we generate the simulated data. Let $\mathbf{c}_i^* \in \mathbb{R}^8$, $i = [1,k]$, be the clustering centers, where the values of the dimensions are 1 or $-1$ with the probability of 1/2. The data in $i$th clustering follows the normal distribution with mean $\mathbf{c}_i^*$ and variance 1. The number of data in each clustering is the same. We use the Gaussian kernel $\kappa(\mathbf{x}, \mathbf{x}') = \exp\left(-\|\mathbf{x}-\mathbf{x}'\|^2/8\right)$.

Generating 10,000 samples for training and 10,000 samples for testing. The number of training samples in each clustering is $10000/k$. The accuracy of kernel $k$-means He & Zhang (2018) on the test set can be written as $\frac{\sum_{i=1}^{\breve{n}} v(\hat{y}, map(y))}{\breve{n}}$, where $y$ is the solution returned by the (approximate)

Table 2: The datasets used in this paper. Test accuracy and training time (in seconds) of kernel $k$-means, Gaussian, Nyström (Liu), Sub-Gaussian sketches, ROS sketches, and Nyström (Ours) on real datasets.

| Dataset | Instance | Class | Kernel $k$-Means | | Gaussian | | Nyström (Liu) | |
|---|---|---|---|---|---|---|---|---|
| | | | Time | Accuracy | Time | Accuracy | Time | Accuracy |
| dna | 2000 | 3 | 0.16 | 0.50±0.01 | 0.12 | 0.49± 0.02 | 0.09 | 0.50±0.02 |
| segment | 2310 | 7 | 0.13 | 0.50±0.02 | 0.09 | 0.45±0.03 | 0.05 | 0.43±0.01 |
| mushrooms | 8124 | 2 | 0.56 | 0.64±0.01 | 0.32 | 0.63±0.02 | 0.11 | 0.61±0.01 |
| pendigits | 10992 | 10 | 0.61 | 0.11 ±0.01 | 0.34 | 0.11±0.01 | 0.21 | 0.10± 0.02 |
| protein | 17766 | 3 | 5.07 | 0.46±0.01 | 3.16 | 0.44±0.03 | 1.09 | 0.45±0.02 |
| a8a | 32561 | 2 | 6.47 | 0.75±0.01 | 3.21 | 0.73±0.03 | 1.12 | 0.73±0.02 |
| w7a | 49749 | 2 | 29.7 | 0.97±0.02 | 15.3 | 0.95±0.02 | 1.36 | 0.96± 0.01 |
| connect-4 | 67557 | 3 | 0.28 | 0.61±0.01 | 0.22 | 0.60±0.03 | 0.11 | 0.59±0.02 |
| covtype | 581012 | 7 | / | / | / | / | / | / |
| Dataset | Instance | Class | Sub-Gaussian (Ours) | | ROS (Ours) | | Nyström (Ours) | |
| | | | Time | Accuracy | Time | Accuracy | Time | Accuracy |
| dna | 2000 | 3 | 0.06 | 0.49± 0.01 | 0.07 | 0.50±0.01 | 0.04 | 0.50±0.01 |
| segment | 2310 | 7 | 0.03 | 0.47±0.03 | 0.03 | 0.49±0.01 | 0.02 | 0.42±0.01 |
| mushrooms | 8124 | 2 | 0.04 | 0.63±0.01 | 0.04 | 0.62±0.02 | 0.03 | 0.60±0.01 |
| pendigits | 10992 | 10 | 0.14 | 0.11±0.01 | 0.16 | 0.11± 0.01 | 0.03 | 0.11±0.02 |
| protein | 17766 | 3 | 0.16 | 0.45±0.01 | 0.21 | 0.46±0.01 | 0.03 | 0.44±0.02 |
| a8a | 32561 | 2 | 0.11 | 0.74±0.01 | 0.12 | 0.74±0.02 | 0.03 | 0.73±0.02 |
| w7a | 49749 | 2 | 0.30 | 0.94±0.02 | 0.36 | 0.95± 0.01 | 0.03 | 0.97±0.01 |
| connect-4 | 67557 | 3 | 0.05 | 0.59±0.01 | 0.06 | 0.60±0.02 | 0.03 | 0.58±0.02 |
| covtype | 581012 | 7 | 1.02 | 0.32±0.02 | 1.36 | 0.33±0.04 | 0.66 | 0.32±0.03 |

kernel $k$-means using Lloyd's algorithm Lloyd (1982), $\hat{y}$ is the real label, and $\ddot{n}$ is the number of data in the test set. If $p = q$, $\upsilon(p, q) = 1$, otherwise $\upsilon(p, q) = 0$. $map(\cdot)$ represents the best mapping to match $\hat{y}$ and $y$. The higher the accuracy, the better the method. The test accuracy and training time of the approximate kernel $k$-means with different $m$ are given in Figure 1 , which can be summarized as follows: (1) There exists a lower bound of the approximate dimensions $m = \sqrt{n} = 100$. When this lower bound is reached, the accuracy of the proposed methods tends to be stable. This is consistent with our theoretical analysis in Theorem 2. (2) The accuracy of the proposed methods keeps the similar accuracy to the exact kernel $k$-means. (3) We take the logarithm of the running time (in seconds) in Figure 1. Our methods (ROS, Sub-Gaussian, Nyström) have obvious advantages over other methods in running time. This verifies our complexity analysis.

## 5.2 Numerical Experiments on Real-World Scenarios

In this subsection, we perform the experiments on the 9 real datasets: dna, segment, mushrooms, pendigits, protein, a8a, w7a, connect-4, and covtype, which are from LIBSVM website [2]. 70 percent of the data in each dataset is used for training experiments, and the rest is used for testing. $m = 150$. The Gaussian kernel is $\exp\left(-\|\mathbf{x} - \mathbf{x}'\|^2/\sigma^2\right)$, where $\sigma = \sqrt{\frac{\sum_{ij} \|x_i - x_j\|^2}{n}}$. The detail of the datasets and experimental results are shown in Table 2. From the above results, we can find that these methods give a similar accuracy as the exact kernel $k$-means. The proposed methods outperform Nyström (Liu) and Gaussian in time cost, which matches our theoretical analysis. If the training time exceeds 90 seconds or the memory is insufficient, the experiment will be stopped. In the large covtype dataset, kernel $k$-means, Gaussian, and Nyström (Liu) cannot achieve the experimental results, but our proposed methods can obtain small training time and good accuracy. Those verify the smaller computational requirements of the proposed methods.

## 6 Conclusions

We propose a unified randomized sketches framework to kernel $k$-means and provide three specific examples of sub-Gaussian sketches, the randomized orthogonal system (ROS) sketches, and Nyström

---

[2]http://www.csie.ntu.edu.tw/~cjlin/libsvm.

kernel $k$-means. Theoretical analysis show that the proposed methods obtain the state-of-the-art risk bound and greatly reduce the computational requirements with sketch dimension $\Omega(\sqrt{n})$. To the best of our knowledge, this is the first optimal excess risk bound with the least approximate dimension and no strong assumptions for general approximate kernel $k$-means. Moreover, we further derive the similarity optimal bound of approximate solutions in the general case, which can be effectively calculated by $k$-means++. The extensive experiments illustrate our theoretical analysis.

## Acknowledgments and Disclosure of Funding

We appreciate all the anonymous reviewers, ACs, and PCs for their invaluable and constructive comments. This work is supported in part by the Special Research Assistant project of CAS (No.E0YY221-2020000702), the National Natural Science Foundation of China (No.62106259, No.62076234), Beijing Outstanding Young Scientist Program (NO.BJJWZYJH012019100020098), and Beijing Natural Science Foundation (No. 4222029). Thank Intelligent Social Governance Platform and Major Innovation & Planning Interdisciplinary Platform for the "Double-First Class" initiative.

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
