# A  Notations and Preliminaries

Let

$$\mathcal{Q}_{\mathbf{C}} := \{q_{\mathbf{C}} = (q_{\mathbf{c}_1}, \ldots, q_{\mathbf{c}_k}) : \mathbf{C} \in \mathcal{H}^k\} \tag{16}$$

be a $k\text{-valued}$ function with $q_{\mathbf{c}_j}(\mathbf{x}) = \|\phi_{\mathbf{x}} - \mathbf{c}_j\|^2$.

**Proposition 2** ($L_\infty$ **Contraction Inequality, Theorem 1 in Foster & Rakhlin (2019)**). *Let $\mathcal{Q} \subseteq \{q : \mathcal{X} \to \mathbb{R}^k\}$ and $l : \mathbb{R}^k \to \mathbb{R}$ be $L$-Lipschitz with respect to the $L_\infty$ norm, that is $\|l(\boldsymbol{v}) - l(\boldsymbol{v}')\|_\infty \le L \cdot \|\boldsymbol{v} - \boldsymbol{v}'\|_\infty, \forall \boldsymbol{v}, \boldsymbol{v}' \in \mathbb{R}^k$. For any $b > 0$, there exists a constant $C > 0$ such that if $\max\{|l(q(\mathbf{x}))|, \|q(\mathbf{x})\|_\infty\} \le \rho$, then*

$$\mathcal{B}_n(l \circ \mathcal{Q}) \le C \cdot L\sqrt{k} \max_i \tilde{\mathcal{B}}_n(\mathcal{Q}_i) \log^{\frac{3}{2}+b}\left(\frac{\rho n}{\max_i \tilde{\mathcal{B}}_n(\mathcal{Q}_i)}\right),$$

*where $\mathcal{B}_n(l \circ \mathcal{Q}) = \mathbb{E}_{\boldsymbol{\sigma}}\left[\sup_{q \in \mathcal{Q}} |\sum_{i=1}^n \sigma_i l(q(\mathbf{x}_i))|\right], \tilde{\mathcal{B}}_n(\mathcal{Q}_i) = \sup_{\mathbf{X} \in \mathcal{X}^n} \mathcal{B}_n(\mathcal{Q}_i).$*

**Proposition 3** (**Lemma 24(a) in Lei et al. (2019)**). *Let $\eta_1, \ldots, \eta_n \in \mathcal{H}$, where $\mathcal{H}$ is a Hilbert space with $\|\cdot\|$ being the associated norm. Let $\sigma_1, \ldots, \sigma_n$ be a sequence of independent Rademacher variables. Then, we have*

$$\mathbb{E}_{\boldsymbol{\sigma}}\left\|\sum_{i=1}^n \sigma_i \eta_i\right\|^2 \le \sum_{i=1}^n \|\eta_i\|^2, \tag{17}$$

*and*

$$\mathbb{E}_{\boldsymbol{\sigma}}\left\|\sum_{i=1}^n \sigma_i \eta_i\right\| \ge \frac{1}{\sqrt{2}}\sqrt{\sum_{i=1}^n \|\eta_i\|^2}. \tag{18}$$

# B  Main Lemmas

To prove the main theorems in this paper, we firstly introduce some lemmas.

**Lemma 2** (**Lemma 3 in Yin et al. (2020c)**). *Let $r_1, r_2$ be any two numbers in $\{+1, -1, 0\}$. For any $a, b \in \mathbb{R}$, let $c = \sqrt{(a^2 + b^2)/2}$. Then $\forall M \in \mathbb{R}$ and $s \in \mathbb{N}_+^0$,*

$$\mathbb{E}\left((M + ar_1 + br_2)^{2s}\right) \le \mathbb{E}\left((M + cr_1 + cr_2)^{2s}\right). \tag{19}$$

**Lemma 3.** *Let $T \sim \mathcal{N}(0, 1)$, $\|\mathbf{k}_i\|^2 \le 1$. $\tilde{\mathbf{k}}_{ij}$ is the element in $i$-th row and $j$-th column of $\tilde{\mathbf{K}}$. For all $s \in \mathbb{N}_+^0$, we have*

$$\mathbb{E}(\tilde{\mathbf{k}}_{ij}^{2s}) \le \mathbb{E}(T^{2s}). \tag{20}$$

*Proof.* Let a "worst-case" unit vector $\mathbf{w} = \frac{1}{\sqrt{n}}(1, \ldots, 1)^T$. For any vector $\mathbf{k}_j$, $\tilde{\mathbf{k}}_{ij} = \mathbf{R}_{\cdot i}\mathbf{k}_j$, where $\mathbf{R}_{\cdot i}$ is the $i$-th row of $\mathbf{R}$.

If $\mathbf{k}_j = (\mathbf{k}_{1j}, \ldots, \mathbf{k}_{nj})^T$ is such that $\mathbf{k}_{ij}^2 = \mathbf{k}_{tj}^2$ for all $i, t$, then by symmetry, $\mathbf{R}_{\cdot i}\mathbf{k}_j$ and $\mathbf{R}_{\cdot i}\mathbf{w}$ are identically distributed and this lemma holds trivially.

Otherwise, we can assume without loss of generality, that $\mathbf{k}_{1j}^2 \neq \mathbf{k}_{2j}^2$ and consider the "more balanced" unit vector $\boldsymbol{\theta} = (c, c, \mathbf{k}_{3j}, \ldots, \mathbf{k}_{nj})^T$, where $c = \sqrt{(\mathbf{k}_{1j}^2 + \mathbf{k}_{2j}^2)/2}$.

We first express $\mathbb{E}(\tilde{\mathbf{k}}_{ij}^{2s})$ as a sum of averages over $r_{i1}, r_{i2}$ and apply Eq.(19) in Lemma 2 to get that each term (average) in the sum, where $r_{i1}$ is the element of $i$-th row and 1-th column of $\mathbf{R}$.

More precisely, in sub-Gaussian case and ROS case,

$$
\begin{aligned}
\mathbb{E}(\tilde{\mathbf{k}}_{ij}^{2s}) &= \mathbb{E}((\mathbf{R}_{\cdot i}\mathbf{k}_j)^{2s}) \\
&= m^{-s} \sum_M \mathbb{E}\left((M + \mathbf{k}_{1j}r_{i1} + \mathbf{k}_{2j}r_{i2})^{2s}\right) \cdot \mathbb{P}\left[\sum_{t=3}^n r_{it}\mathbf{k}_{tj} = \frac{M}{\sqrt{m}}\right] \\
&\leq m^{-s} \sum_M \mathbb{E}\left((M + cr_{i1} + cr_{i2})^{2s}\right) \cdot \mathbb{P}\left[\sum_{t=3}^n r_{it}\mathbf{k}_{tj} = \frac{M}{\sqrt{m}}\right] \\
&= \mathbb{E}((\mathbf{R}_{\cdot i}\boldsymbol{\theta})^{2s}).
\end{aligned}
$$

In Nyström case,

$$
\begin{aligned}
\mathbb{E}(\tilde{\mathbf{k}}_{ij}^{2s}) &= \mathbb{E}((\mathbf{R}_{\cdot i}\mathbf{k}_j)^{2s}) \\
&= \sum_M \mathbb{E}\left((M + \mathbf{k}_{1j}r_{i1} + \mathbf{k}_{2j}r_{i2})^{2s}\right) \cdot \mathbb{P}\left[\sum_{t=3}^n r_{it}\mathbf{k}_{tj} = M\right] \\
&\leq \sum_M \mathbb{E}\left((M + cr_{i1} + cr_{i2})^{2s}\right) \cdot \mathbb{P}\left[\sum_{t=3}^n r_{it}\mathbf{k}_{tj} = M\right] \\
&= \mathbb{E}((\mathbf{R}_{\cdot i}\boldsymbol{\theta})^{2s}).
\end{aligned}
$$

Applying this argument repeatedly yields the lemma, as $\boldsymbol{\theta}$ eventually becomes $\mathbf{w}$, we obtain

$$
\mathbb{E}(\tilde{\mathbf{k}}_{ij}^{2s}) = \mathbb{E}((\mathbf{R}_{\cdot i}\mathbf{k}_j)^{2s}) \leq \mathbb{E}((\mathbf{R}_{\cdot i}\mathbf{w})^{2s}). \tag{21}
$$

In the following, we prove $\mathbb{E}((\mathbf{R}_{\cdot i}\mathbf{w})^{2s}) \leq \mathbb{E}(T^{2s})$.

To simplify notation, we write $r_{it} = Y_t$. Thus, in sub-Gaussian case and ROS case, $\mathbf{R}_{\cdot i}\mathbf{w} = \frac{1}{\sqrt{nm}}\sum_{t=1}^n Y_t$. In Nyström case, $\mathbf{R}_{\cdot i}\mathbf{w} = \frac{1}{\sqrt{n}}\sum_{t=1}^n Y_t$.

Let $\{T_i\}_{i=1}^n$ be a family of i.i.d. standard Normal random variables. Then $\sum_{i=1}^n T_i$ is a Normal random variable with variance $n$. Therefore, $T = \frac{1}{\sqrt{n}}\sum_{i=1}^n T_i$ and $T \sim \mathcal{N}(0, 1)$.

For every $s = 0, 1, \ldots,$

$$
\mathbb{E}(T^{2s}) = \frac{1}{(\sqrt{n})^{2s}} \sum_{i_1=1}^n \cdots \sum_{i_{2s}=1}^n \mathbb{E}(T_{i_1} \cdots T_{i_{2s}}), \tag{22}
$$

and in sub-Gaussian and ROS cases

$$
\mathbb{E}((\mathbf{R}_{\cdot i}\mathbf{w})^{2s}) = \frac{1}{(\sqrt{nm})^{2s}} \sum_{i_1=1}^n \cdots \sum_{i_{2s}=1}^n \mathbb{E}(Y_{i_1} \cdots Y_{i_{2s}}),
$$

in Nyström case

$$
\mathbb{E}((\mathbf{R}_{\cdot i}\mathbf{w})^{2s}) = \frac{1}{(\sqrt{n})^{2s}} \sum_{i_1=1}^n \cdots \sum_{i_{2s}=1}^n \mathbb{E}(Y_{i_1} \cdots Y_{i_{2s}}).
$$

To prove this lemma, in the following, we will prove that for every value assignment to the indices $i_1, \ldots, i_{2s}$,

$$\mathbb{E}(Y_{i_1} \cdots Y_{i_{2s}}) \leq \mathbb{E}(T_{i_1} \cdots T_{i_{2s}}). \tag{23}$$

In sub-Gaussian case:

Let $V = \langle v_1, v_2, \ldots, v_{2s} \rangle$ be the value assignment considered. For $i \in \{1, \ldots, n\}$, let $c_V(i)$ be the number of times that $i$ appears in $V$. Observe that if for some $i$, $c_V(i)$ is odd then the expectations appearing in Eq.(22) are 0, since $\{Y_i\}_{i=1}^n$ and $\{T_i\}_{i=1}^n$ are independent families and $\mathbb{E}(Y_i) = \mathbb{E}(T_i) = 0$ for all $i$. Thus, we can assume that there exists a set $\{j_1, j_2, \ldots, j_p\}$ of indices and corresponding values $\{l_1, l_2, \ldots, l_p\}$ such that

$$\mathbb{E}(T_{i_1} \cdots T_{i_{2s}}) = \mathbb{E}(T_{j_1}^{2l_1} T_{j_2}^{2l_2} \cdots T_{j_p}^{2l_p})$$

and

$$\mathbb{E}(Y_{i_1} \cdots Y_{i_{2s}}) = \mathbb{E}(Y_{j_1}^{2l_1} Y_{j_2}^{2l_2} \cdots Y_{j_p}^{2l_p}).$$

Note that since the indices $j_1, j_2, \cdots, j_p$ are distinct, $\{T_{j_t}\}_{t=1}^p$ and $\{Y_{j_t}\}_{t=1}^p$ are families of i.i.d. Therefore,

$$\mathbb{E}(T_{i_1} \cdots T_{i_{2s}}) = \mathbb{E}(T_{j_1}^{2l_1}) \times \cdots \times \mathbb{E}(T_{j_p}^{2l_p}) \tag{24}$$

and

$$\mathbb{E}(Y_{i_1} \cdots Y_{i_{2s}}) = \mathbb{E}(Y_{j_1}^{2l_1}) \times \cdots \times \mathbb{E}(Y_{j_p}^{2l_p}).$$

So, in order to prove Eq.(23) it suffices to prove that for every $l = 0, 1, \ldots$

$$\mathbb{E}(Y_1^{2l}) \leq \mathbb{E}(T_1^{2l}).$$

We know that $(2l)$-th moment of $\mathcal{N}(0, 1)$ is

$$(2l - 1)!! = (2l)!/(l!2^l) \geq 1. \tag{25}$$

For all $l \geq 0$, we have $\mathbb{E}(Y_1^{2l}) \leq 1$. Therefore, we have $\mathbb{E}(Y_{i_1} \cdots Y_{i_{2s}}) \leq \mathbb{E}(T_{i_1} \cdots T_{i_{2s}})$.

In ROS and Nyström cases:

$\{Y_i\}_{i=1}^n$ is a family of i.i.d. One knows that

$$\mathbb{E}(Y_{i_1} \cdots Y_{i_{2s}}) = \mathbb{E}(Y_{i_1}) \times \cdots \times \mathbb{E}(Y_{i_{2s}}).$$

Combining $-1 \leq \mathbb{E}(Y_{i_1}) \leq 1$, Eq.(24), and Eq.(25), we know that $\mathbb{E}(Y_{i_1} \cdots Y_{i_{2s}}) \leq \mathbb{E}(T_{i_1} \cdots T_{i_{2s}})$. Here, we complete the proof of $\mathbb{E}(Y_{i_1} \cdots Y_{i_{2s}}) \leq \mathbb{E}(T_{i_1} \cdots T_{i_{2s}})$ and $\mathbb{E}((\mathbf{R}_{\cdot i} \mathbf{w})^{2s}) \leq \mathbb{E}(T^{2s})$.

Combining $\mathbb{E}((\mathbf{R}_{\cdot i} \mathbf{w})^{2s}) \leq \mathbb{E}(T^{2s})$ and Eq.(21), we obtain $\mathbb{E}(\tilde{\mathbf{k}}_{ij}^{2s}) \leq \mathbb{E}(T^{2s})$. $\qquad\square$

**Lemma 4.** *For all $h \in [0, m/2)$, and $\|\mathbf{k}_i\|^2 \leq 1$, we have*

$$\mathbb{E}(\exp(h\tilde{\mathbf{k}}_{ij}^2)) \leq \frac{1}{\sqrt{1 - 2h/m}}, \tag{26}$$

*and*

$$\mathbb{E}(\tilde{\mathbf{k}}_{ij}^4) \leq 3/m^2. \tag{27}$$

*Proof.* According to Lemma 3, we know

$$\mathbb{E}(\tilde{\mathbf{k}}_{ij}^4) \leq \mathbb{E}(T^4), \tag{28}$$

while

$$\mathbb{E}(T^4) = \int_{-\infty}^{+\infty} \frac{1}{\sqrt{2\pi}} \exp(-\lambda^2/2) \left(\lambda^4/m^2\right) m\lambda = 3/m^2.$$

The following will prove Eq.(26).

For any real-valued random variable $U$ and for all $h$ such that $\mathbb{E}(\exp(hU^2))$ is bounded. According to the Monotone Convergence Theorem (MCT), we get the formula

$$\mathbb{E}(\exp(hU^2)) = \mathbb{E}\left(\sum_{t=0}^{\infty} \frac{(hU^2)^t}{t!}\right) = \sum_{t=0}^{\infty} \frac{h^t}{t!}\mathbb{E}(U^{2t}).$$

Here we obtain

$$
\begin{aligned}
\mathbb{E}(\exp(hT^2)) &= \int_{-\infty}^{+\infty} \frac{1}{\sqrt{2\pi}} \exp(-\lambda^2/2)\exp(h\lambda^2/m)m\lambda \\
&= \frac{1}{\sqrt{1-2h/m}} = \sum_{t=0}^{\infty} \frac{h^t}{t!}\mathbb{E}(T^{2t}) \\
&\geq \sum_{t=0}^{\infty} \frac{h^t}{t!}\mathbb{E}(\tilde{\mathbf{k}}_{ij}^{2t}) = \mathbb{E}(\exp(h\tilde{\mathbf{k}}_{ij}^2)).
\end{aligned}
\tag{29}
$$

For converge, we take $h \in [0, m/2)$ and apply the MCT in Eq.(38). Therefore, we have $\mathbb{E}(\exp(h\tilde{\mathbf{k}}_{ij}^2)) \leq \frac{1}{\sqrt{1-2h/m}}$, for $h \in [0, m/2)$. This proof logic is similar to Yin et al. (2020c). $\qquad\square$

**Lemma 5.** *Let $\mathcal{S}$ be an arbitrary set of $n$ samples in $\mathcal{X}$ and $\mathbf{K} \in \mathbb{R}^{n\times n}$ be its kernel matrix. The $i$-th column of $\mathbf{K}$ is represented by $\mathbf{k}_i$. Given $\varepsilon, \delta \in (0, 1)$, let*

$$m = \Omega\left(\frac{4\log n - 2\log \delta}{\varepsilon - \log(1+\varepsilon)}\right),\tag{30}$$

$$\tilde{\mathbf{K}} = \mathbf{R}\mathbf{K} \in \mathbb{R}^{m\times n},$$

*and $\mathbf{R}$ be a $m \times n$ random matrix in one of the three cases of sub-Gaussian, ROS, and Nyström. And let $f : \mathbb{R}^n \to \mathbb{R}^m$ map the $i$-th column of $\mathbf{K}$ to the $i$-th column of $\tilde{\mathbf{K}}$.*

*For all $\mathbf{k}_i, \mathbf{k}_j \in \mathbf{K}$, with probability at least $1 - \delta$, we have,*
$$(1-\varepsilon)\|\mathbf{k}_i - \mathbf{k}_j\|^2 \leq \|f(\mathbf{k}_i) - f(\mathbf{k}_j)\|^2 \leq (1+\varepsilon)\|\mathbf{k}_i - \mathbf{k}_j\|^2.$$

*Proof.* For arbitrary $h > 0$, according to Markov's inequality we get

$$
\begin{aligned}
\mathbb{P}\left[\frac{\sum_{i=1}^m (\mathbf{R}_{\cdot i} \cdot \mathbf{k}_j^T)^2}{\|\mathbf{k}_j\|^2} > 1+\varepsilon\right] &= \mathbb{P}\left[\exp(h\frac{\sum_{i=1}^m (\mathbf{R}_{\cdot i} \cdot \mathbf{k}_j^T)^2}{\|\mathbf{k}_j\|^2}) > \exp(h(1+\varepsilon))\right] \\
&< \mathbb{E}\left(\exp(h\frac{\sum_{i=1}^m (\mathbf{R}_{\cdot i} \cdot \mathbf{k}_j^T)^2}{\|\mathbf{k}_j\|^2})\right)\exp\left(-h(1+\varepsilon)\right).
\end{aligned}
\tag{31}
$$

Let $\|\mathbf{k}_1\|^2 = 1$, we have:

$$
\begin{aligned}
\mathbb{E}\left(\exp(h\frac{\sum_{i=1}^m (\mathbf{R}_{\cdot i} \cdot \mathbf{k}_j^T)^2}{\|\mathbf{k}_j\|^2})\right) &= \mathbb{E}\left(\prod_{i=1}^m \exp(h\frac{(\mathbf{R}_{\cdot i} \cdot \mathbf{k}_j^T)^2}{\|\mathbf{k}_j\|^2})\right) \\
&= \left(\mathbb{E}\left(\exp(h\frac{(\mathbf{R}_{\cdot i} \cdot \mathbf{k}_1^T)^2}{\|\mathbf{k}_1\|^2})\right)\right)^m \\
&= \left(\mathbb{E}\left(\exp(h(\mathbf{R}_{\cdot i} \cdot \mathbf{k}_1^T)^2)\right)\right)^m.
\end{aligned}
\tag{32}
$$

According to Eq.(26) of Lemma 4, we have

$$\mathbb{E}\left(\exp(h(\mathbf{R}_{\cdot i} \cdot \mathbf{k}_1^T)^2)\right) \leq \frac{1}{\sqrt{1-2h/m}}.\tag{33}$$

Let $h = \frac{m\varepsilon}{2(1+\varepsilon)} < \frac{m}{2}$. Taking Eq.(31), Eq.(32), and Eq.(33) to Eq.(34), for any $0 < \varepsilon < 1$, we obtain that

$$
\begin{aligned}
\mathbb{P}\left[\frac{\sum_{i=1}^m (\mathbf{R}_{\cdot i} \cdot \mathbf{k}_j^T)^2}{\|\mathbf{k}_j\|^2} > 1+\varepsilon\right] &< \left(\mathbb{E}\left(\exp(h(\mathbf{R}_{\cdot i} \cdot \mathbf{k}_1^T)^2)\right)\right)^m \exp\left(-h(1+\varepsilon)\right) \\
&\leq \left(\frac{1}{\sqrt{1-2h/m}}\right)^m \exp\left(-h(1+\varepsilon)\right) \\
&= \left(\frac{1}{1+\varepsilon}\right)^{-m/2} \exp\left(\frac{-m\varepsilon}{2}\right).
\end{aligned}
\tag{34}
$$

Similarly, for arbitrary $h > 0$ and $0 < \varepsilon < 1$, we have

$$\mathbb{P}\Big[\frac{\sum_{i=1}^m (\mathbf{R}_{\cdot i} \cdot \mathbf{k}_j^T)^2}{\|\mathbf{k}_j\|^2} < 1 - \varepsilon\Big] = \mathbb{P}\Big[\exp(h\frac{\sum_{i=1}^m (\mathbf{R}_{\cdot i} \cdot \mathbf{k}_j^T)^2}{\|\mathbf{k}_j\|^2}) < \exp(h(1-\varepsilon))\Big]$$

$$< \mathbb{E}\Big(\exp(-h\frac{\sum_{i=1}^m (\mathbf{R}_{\cdot i} \cdot \mathbf{k}_j^T)^2}{\|\mathbf{k}_j\|^2})\Big) \exp\Big(h(1-\varepsilon)\Big) \tag{35}$$

$$= \big(\mathbb{E}\left(\exp(-h(\mathbf{R}_{\cdot i} \cdot \mathbf{k}_1^T)^2)\right)\big)^m \exp\Big(h(1-\varepsilon)\Big).$$

By expanding $\exp(-h(\mathbf{R}_{\cdot i} \cdot \mathbf{k}_1^T)^2)$, we get that

$$\mathbb{P}\Big[\frac{\sum_{i=1}^m (\mathbf{R}_{\cdot i} \cdot \mathbf{k}_j^T)^2}{\|\mathbf{k}_j\|^2} < 1 - \varepsilon\Big]$$

$$< \Big(\mathbb{E}\Big(1 - h(\mathbf{R}_{\cdot i} \cdot \mathbf{k}_1^T)^2 + \frac{(-h(\mathbf{R}_{\cdot i} \cdot \mathbf{k}_1^T)^2)^2}{2!}\Big)\Big)^m \exp\Big(h(1-\varepsilon)\Big) \tag{36}$$

$$= \Big(1 - h\mathbb{E}((\mathbf{R}_{\cdot i} \cdot \mathbf{k}_1^T)^2) + \frac{h^2}{2}\mathbb{E}((\mathbf{R}_{\cdot i} \cdot \mathbf{k}_1^T)^4)\Big)^m \exp\Big(h(1-\varepsilon)\Big).$$

According to Eq.(27) in Lemma 4, we know

$$\mathbb{E}((\mathbf{R}_{\cdot i} \cdot \mathbf{k}_1^T)^4) \leq 3/m^2. \tag{37}$$

According to Eq.(38), we have

$$\frac{1}{\sqrt{1-2h/m}} \geq \sum_{t=0}^{\infty} \frac{h^t}{t!}\mathbb{E}(\tilde{\mathbf{k}}_{ij}^{2t}) \geq \sum_{t=0}^{1} \frac{h^t}{t!}\mathbb{E}(\tilde{\mathbf{k}}_{ij}^{2t}) = 1 + h\mathbb{E}(\tilde{\mathbf{k}}_{ij}^2) = 1 + h\mathbb{E}((\mathbf{R}_{\cdot i} \cdot \mathbf{k}_1^T)^2). \tag{38}$$

So, one can obtain

$$\mathbb{E}((\mathbf{R}_{\cdot i} \cdot \mathbf{k}_1^T)^2) \leq \frac{1}{h}\Big(\frac{1}{\sqrt{1-2h/m}} - 1\Big). \tag{39}$$

Let $h = \frac{m}{2} \cdot \frac{\varepsilon}{(1+\varepsilon)} < \frac{m}{2}$. Taking Eq.(37) and Eq.(39) into Eq.(36), we obtain that

$$\mathbb{P}\Big[\frac{\sum_{i=1}^m (\mathbf{R}_{\cdot i} \cdot \mathbf{k}_j^T)^2}{\|\mathbf{k}_j\|^2} < 1 - \varepsilon\Big]$$

$$< \Big(1 - \big(\frac{1}{\sqrt{1-2h/m}} - 1\big) + \frac{3h^2}{2m^2}\Big)^m \exp\Big(h(1-\varepsilon)\Big) \tag{40}$$

$$< \Big(\frac{1}{1+\varepsilon}\Big)^{-m/2} \exp\Big(\frac{-m\varepsilon}{2}\Big).$$

Let

$$2 \times \Big(\frac{1}{1+\varepsilon}\Big)^{-m/2} \exp\Big(\frac{-m\varepsilon}{2}\Big) \leq 2\delta/n^2,$$

we obtain

$$m \geq \frac{4\log n - 2\log \delta}{\varepsilon - \log(1+\varepsilon)}.$$

According to Eq.(34) and Eq.(40), we know that, for each of the $\binom{n}{2}$ pairs $\mathbf{k}_i, \mathbf{k}_j$, with the probability of $1 - \binom{n}{2} \times 2\delta/n^2 > 1 - \delta$, the squared norm of the vector $\mathbf{k}_i - \mathbf{k}_j$ is maintained within a factor of $1 \pm \varepsilon$. That is, with the probability at least $1 - \delta$, for all $\mathbf{k}_i, \mathbf{k}_j \in \mathbf{K}$,

$$(1-\varepsilon)\|\mathbf{k}_i - \mathbf{k}_j\|^2 \leq \|\tilde{\mathbf{k}}_i - \tilde{\mathbf{k}}_j\|^2 \leq (1+\varepsilon)\|\mathbf{k}_i - \mathbf{k}_j\|^2. \tag{41}$$

$\square$

**Lemma 6.** *If constructing the random matrix $\mathbf{R}$ in one of the three cases of sub-Gaussian, ROS, and Nyström, by*

$$m = \Omega\left(\frac{4\log n - 2\log\delta}{\varepsilon - \log(1+\varepsilon)}\right),$$

*given any $\varepsilon, \delta \in (0,1)$, then we have, with probability at least $1 - \delta$,*

$$W(\tilde{\mathbf{C}}_{n,m}, \mu_n) - W(\mathbf{C}_n, \mu_n) \leq \frac{2\varepsilon}{1-\varepsilon}. \tag{42}$$

*Proof.* We denote by $\bar{\mathbf{C}}_{n,m} = [\bar{\mathbf{c}}_1, \ldots, \bar{\mathbf{c}}_k]$ the empirical clustering centers associated with the $m$-dimensional embeddings $\tilde{\mathbf{k}}_1, \ldots, \tilde{\mathbf{k}}_n$. Each $\bar{\mathbf{c}}_j$ is the mean of those $\tilde{\mathbf{k}}_i$'s in the Voronoi cell $\tilde{\mathcal{C}}_j$, that is

$$\bar{\mathbf{c}}_j = \frac{\sum_{i=1}^n \tilde{\mathbf{k}}_i \mathbb{I}_{\{\tilde{\mathbf{k}}_i \in \tilde{\mathcal{C}}_j\}}}{\sum_{i=1}^n \mathbb{I}_{\{\tilde{\mathbf{k}}_i \in \tilde{\mathcal{C}}_j\}}}, \quad j = 1, \ldots, k.$$

Let $\tilde{\alpha}_j = \sum_{i=1}^n \mathbb{I}_{\{\tilde{\mathbf{k}}_i \in \tilde{\mathcal{C}}_j\}}$ and $\beta_j = \sum_{i=1}^n \mathbb{I}_{\{\mathbf{k}_i \in \mathcal{C}_j\}}$. We have

$$\begin{aligned}
W\left(\bar{\mathbf{C}}_{n,m}, \mu_n\right) &= \frac{1}{n}\sum_{i=1}^n \min_{j=[k]} \left\|\tilde{\mathbf{k}}_i - \bar{\mathbf{c}}_j\right\|^2 \\
&= \frac{1}{n}\sum_{j=1}^k \sum_{i=1}^n \left\|\tilde{\mathbf{k}}_i - \bar{\mathbf{c}}_j\right\|^2 \mathbb{I}_{\{\tilde{\mathbf{k}}_i \in \tilde{\mathcal{C}}_j\}} \\
&= \sum_{j=1}^k \frac{1}{2n\tilde{\alpha}_j} \sum_{i_1, i_2=1}^n \left\|\tilde{\mathbf{k}}_{i_1} - \tilde{\mathbf{k}}_{i_2}\right\|^2 \mathbb{I}_{\left\{(\tilde{\mathbf{k}}_{i_1}, \bar{\mathbf{k}}_{i_2}) \in \tilde{\mathcal{C}}_j^2\right\}}.
\end{aligned}$$

Combining the optimality of the $k$-means procedure (Lemma 1 in Linder (2002)), we get

$$W\left(\bar{\mathbf{C}}_{n,m}, \mu_n\right) \leq \sum_{j=1}^k \frac{1}{2n\beta_j} \sum_{i_1,i_2=1}^n \left\|\tilde{\mathbf{k}}_{i_1} - \tilde{\mathbf{k}}_{i_2}\right\|^2 \mathbb{I}_{\left\{(\mathbf{k}_{i_1}, \mathbf{k}_{i_2}) \in \mathcal{C}_j^2\right\}}.$$

Therefore, combining Lemma 5, with probability at least $1 - \delta$, we have

$$\begin{aligned}
W\left(\bar{\mathbf{C}}_{n,m}, \mu_n\right) &\leq (1+\varepsilon)\sum_{j=1}^k \frac{1}{2n\beta_j} \sum_{i_1,i_2=1}^n \|\mathbf{k}_{i_1} - \mathbf{k}_{i_2}\|^2 \mathbb{I}_{\left\{(\mathbf{k}_{i_1}, \mathbf{k}_{i_2}) \in \mathcal{C}_j^2\right\}} \\
&= (1+\varepsilon)W\left(\mathbf{C}_n, \mu_n\right).
\end{aligned}$$

Using the similar proof methods, we can obtain

$$(1-\varepsilon)W(\tilde{\mathbf{C}}_{n,m}, \mu_n) \leq W\left(\bar{\mathbf{C}}_{n,m}, \mu_n\right).$$

Note that $W(\mathbf{c}_n, \mu_n) \leq 1$ and $\varepsilon \in (0,1)$. So, we have

$$W(\tilde{\mathbf{C}}_{n,m}, \mu_n) - W(\mathbf{C}_n, \mu_n) \leq \frac{2\varepsilon}{1-\varepsilon}W(\mathbf{C}_n, \mu_n) \leq \frac{2\varepsilon}{1-\varepsilon}.$$

$\square$

**Lemma 7.** *For $\delta \in (0,1)$, with probability $1 - \delta$, we have*

$$\sup_{q_{\mathbf{C}} \in \mathcal{Q}_{\mathbf{C}}} \left|\sum_{i=1}^n \sigma_i q_{\mathbf{C}}(\mathbf{x})\right| \leq \mathcal{O}\left(\sqrt{kn}\log^2(\sqrt{n})\right). \tag{43}$$

*Proof.* Note that $\|\phi_{\mathbf{x}}\| \leq 1$ and $\mathcal{Q}_{\mathbf{C}} := \{q_{\mathbf{C}} = (q_{\mathbf{c}_1}, \ldots, q_{\mathbf{c}_k}) : \mathbf{C} \in \mathcal{H}^k\}$ is a *k-valued* function with $q_{\mathbf{c}_j}(\mathbf{x}) = \|\phi_{\mathbf{x}} - \mathbf{c}_j\|^2$. Therefore, we have $\|\mathbf{c}_j\| \leq 1$, $q_{\mathbf{c}_j}(\mathbf{x}) \leq 2\|\phi_{\mathbf{x}}\| + 2\|\mathbf{c}_j\| \leq 4$, $\|q_{\mathbf{C}}(\mathbf{x})\|_\infty = \max_j |q_{\mathbf{c}_j}(\mathbf{x})| \leq 4$, and $|l(q_{\mathbf{C}}(\mathbf{x}))| = |\min_{j=[k]} q_{\mathbf{c}_j}(\mathbf{x})| \leq 4$, for all $\mathbf{x} \in \mathcal{X}$.

Due to $q_{\mathbf{c}_j}(\mathbf{x}) \leq 2\|\phi_{\mathbf{x}}\| + 2\|\mathbf{c}_j\| \leq 4$, we get

$$\max \left\{ \sup_{\mathbf{x} \in \mathcal{X}} \sup_{q_{\mathbf{C}} \in \mathcal{Q}_{\mathbf{C}_i}} |q_{\mathbf{C}}(\mathbf{x})|, i = 1, \ldots, k \right\} \leq 4. \tag{44}$$

According to $L_\infty$ contraction inequality in Proposition 2, with $L = 1, \rho = 4$, and $b = 1/2$, one can get

$$\sup_{q_{\mathbf{C}} \in \mathcal{Q}_{\mathbf{C}}} \left| \sum_{i=1}^n \sigma_i q_{\mathbf{C}}(\mathbf{x}) \right| \leq C \cdot \sqrt{k} \max_i \tilde{\mathcal{B}}_n \left( \mathcal{Q}_{\mathbf{C}_i} \right) \log^2 \left( \frac{4n}{\max_i \tilde{\mathcal{B}}_n \left( \mathcal{Q}_{\mathbf{C}_i} \right)} \right), \tag{45}$$

where $\tilde{\mathcal{B}}_n \left( \mathcal{Q}_{\mathbf{C}_i} \right) = \sup_{\mathbf{X} \in \mathcal{X}^n} \mathcal{B}_n \left( \mathcal{Q}_{\mathbf{C}_i} \right), \mathcal{B}_n \left( \mathcal{Q}_{\mathbf{C}} \right) = \sup_{q_{\mathbf{C}} \in \mathcal{Q}_{\mathbf{C}}} |\sum_{i=1}^n \sigma_i q_{\mathbf{C}}(\mathbf{x})|$, and $C$ is a constant.

For all $j$, we get,

$$\tilde{\mathcal{B}}_n \left( \mathcal{Q}_{\mathbf{C}_j} \right) = \sup_{\mathbf{X} \in \mathcal{X}^n} \mathbb{E}_{\boldsymbol{\sigma}} \left[ \sup_{q_{\mathbf{C}} \in \mathcal{Q}_{\mathbf{C}_j}} \left| \sum_{i=1}^n \sigma_i q_{\mathbf{C}}(\mathbf{x}_i) \right| \right]$$

$$\geq \sup_{\mathbf{x} \in \mathcal{X}} \mathbb{E}_{\boldsymbol{\sigma}} \left[ \sup_{q_{\mathbf{C}} \in \mathcal{Q}_{\mathbf{C}_j}} \left| \sum_{i=1}^n \sigma_i q_{\mathbf{C}}(\mathbf{x}) \right| \right]$$

$$\geq \sup_{\mathbf{x} \in \mathcal{X}, q_{\mathbf{C}} \in \mathcal{Q}_{\mathbf{C}_j}} \mathbb{E}_{\boldsymbol{\sigma}} \left| \sum_{i=1}^n \sigma_i q_{\mathbf{C}}(\mathbf{x}) \right| \tag{46}$$

$$\geq \frac{\sqrt{n}}{\sqrt{2}} \sup_{\mathbf{x} \in \mathcal{X}, q_{\mathbf{C}} \in \mathcal{Q}_{\mathbf{C}_j}} \sqrt{|q_{\mathbf{C}}(\mathbf{x})|}. \tag{47}$$

According to Jensen's inequality, we obtain Eq.(46). Eq.(47) is obtained by Eq.(18) of Proposition 3. So, we have

$$\max_i \tilde{\mathcal{B}}_n \left( \mathcal{Q}_{\mathbf{C}_i} \right) \geq \frac{\sqrt{n} \sqrt{\max \left\{ \sup_{\mathbf{x} \in \mathcal{X}} \sup_{q_{\mathbf{C}} \in \mathcal{Q}_{\mathbf{C}_i}} |q_{\mathbf{C}}(\mathbf{x})|, i = 1, \ldots, k \right\}}}{\sqrt{2}}. \tag{48}$$

For $i \in \{1, \ldots, k\}$, we have

$$\mathbb{E}_{\boldsymbol{\sigma}} \sup_{q_{\mathbf{C}}} \in \mathcal{Q}_{\mathbf{C}_i} \left| \sum_{j=1}^n \sigma_j q_{\mathbf{C}}(\mathbf{x}_j) \right| = \mathbb{E}_{\boldsymbol{\sigma}} \sup_{\mathbf{c} \in \mathcal{H}} \left| \sum_{j=1}^n \sigma_j \|\phi_j - \mathbf{c}\|^2 \right|$$

$$\leq 2 \mathbb{E}_{\boldsymbol{\sigma}} \sup_{\mathbf{c} \in \mathcal{H}} \left| \sum_{j=1}^n \sigma_j \langle \phi_j, \mathbf{c} \rangle \right| + \mathbb{E}_{\boldsymbol{\sigma}} \sup_{\mathbf{c} \in \mathcal{H}} \left| \sum_{j=1}^n \sigma_j \|\mathbf{c}\|^2 \right| \tag{49}$$

According to Eq.(17) of Proposition 3 and $\|\mathbf{c}\| \leq 1$, we have

$$\mathbb{E}_{\boldsymbol{\sigma}} \sup_{\mathbf{c} \in \mathcal{H}} \left| \sum_{j=1}^n \sigma_j \|\mathbf{c}\|^2 \right| \leq \mathbb{E}_{\boldsymbol{\sigma}} \left| \sum_{j=1}^n \sigma_j \right| \leq \sqrt{\mathbb{E}_{\boldsymbol{\sigma}} \left| \sum_{j=1}^n \sigma_j \right|^2} \leq \sqrt{n}, \tag{50}$$

and

$$\mathbb{E}_{\boldsymbol{\sigma}} \sup_{\mathbf{c} \in \mathcal{H}} \left| \sum_{j=1}^n \sigma_j \langle \phi_j, \mathbf{c} \rangle \right| = \mathbb{E}_{\boldsymbol{\sigma}} \sup_{\mathbf{c} \in \mathcal{H}} \left| \left\langle \sum_{j=1}^n \sigma_j \phi_j, \mathbf{c} \right\rangle \right| \leq \sqrt{\mathbb{E}_{\boldsymbol{\sigma}} \left\| \sum_{j=1}^n \sigma_j \phi_j \right\|^2}$$

$$\leq \sqrt{\sum_{i=1}^n \|\phi_i\|^2} \leq \sqrt{n}. \tag{51}$$

Combining Eq.(49), Eq.(50) and Eq.(51), we obtain

$$\max_i \tilde{\mathcal{B}}_n \left( \mathcal{Q}_{\mathbf{C}_i} \right) \leq 3\sqrt{n}. \tag{52}$$

Combining Eq.(44), Eq.(45), Eq.(48), and Eq.(52), we have

$$\sup_{q_{\mathbf{C}} \in \mathcal{Q}_{\mathbf{C}}} \left| \sum_{i=1}^{n} \sigma_i q_{\mathbf{C}}(\mathbf{x}) \right| \leq 3C_1 \sqrt{kn} \log^2(\sqrt{n}), \tag{53}$$

where $C_1$ is a constant. Here we complete this proof. $\qquad\square$

**Lemma 8.** *For $\delta \in (0,1)$, with probability $1 - \delta$, we have*

$$\mathbb{E}\left[ W(\tilde{\mathbf{C}}_{n,m}, \mu) - W(\tilde{\mathbf{C}}_{n,m}, \mu_n) \right] \leq \mathcal{O}\left( \frac{\sqrt{k} \log^2(\sqrt{n}) + \sqrt{\log \frac{1}{\delta}}}{\sqrt{n}} \right). \tag{54}$$

*Proof.* Let $\mathbf{x}'_1, \ldots, \mathbf{x}'_n$ be a copy of $\mathbf{x}_1, \ldots, \mathbf{x}_n$, independent of the $\sigma_i$'s. According to a standard symmetrization argument Bartlett & Mendelson (2002), one can obtain that

$$
\begin{aligned}
&\mathbb{E} \sup_{\mathbf{C} \in \mathcal{H}^k} |W(\mathbf{C}, \mu) - W(\mathbf{C}, \mu_n)| \\
&\leq \mathbb{E} \sup_{q_{\mathbf{C}} \in \mathcal{Q}_{\mathbf{C}}} \left| \frac{1}{n} \sum_{i=1}^{n} \sigma_i \left[ q_{\mathbf{C}}(\mathbf{x}) - q_{\mathbf{C}}\left(\mathbf{x}'\right) \right] \right| \\
&\leq 2\mathbb{E} \sup_{q_{\mathbf{C}} \in \mathcal{Q}_{\mathbf{C}}} \left| \frac{1}{n} \sum_{i=1}^{n} \sigma_i q_{\mathbf{C}}(\mathbf{x}) \right| = \frac{2}{n} \mathbb{E} \sup_{q_{\mathbf{C}} \in \mathcal{Q}_{\mathbf{C}}} \left| \sum_{i=1}^{n} \sigma_i q_{\mathbf{C}}(\mathbf{x}) \right|.
\end{aligned}
\tag{55}
$$

According to Bartlett & Mendelson (2002), we have, with probability $1 - \delta$,

$$\mathbb{E} \sup_{q_{\mathbf{C}} \in \mathcal{Q}_{\mathbf{C}}} \left| \sum_{i=1}^{n} \sigma_i q_{\mathbf{C}}(\mathbf{x}) \right| \leq \sup_{q_{\mathbf{C}} \in \mathcal{Q}_{\mathbf{C}}} \left| \sum_{i=1}^{n} \sigma_i q_{\mathbf{C}}(\mathbf{x}) \right| + \sqrt{2n \log \frac{1}{\delta}}. \tag{56}$$

According to Lemma 7, we know, with probability $1 - \delta$, $\sup_{q_{\mathbf{C}} \in \mathcal{Q}_{\mathbf{C}}} |\sum_{i=1}^{n} \sigma_i q_{\mathbf{C}}(\mathbf{x})| \leq \mathcal{O}\left(\sqrt{kn} \log^2(\sqrt{n})\right)$. Therefore, combining Eq.(55), Eq.(56), and Lemma 7, one can obtain

$$\mathbb{E} \sup_{\mathbf{C} \in \mathcal{H}^k} |W(\mathbf{C}, \mu_n) - W(\mathbf{C}, \mu)| \leq \mathcal{O}\left( \frac{\sqrt{k} \log^2(\sqrt{n}) + \sqrt{\log \frac{1}{\delta}}}{\sqrt{n}} \right). \tag{57}$$

Note that $\mathbb{E}\left[ W(\tilde{\mathbf{C}}_{n,m}, \mu) - W(\tilde{\mathbf{C}}_{n,m}, \mu_n) \right] \leq \mathbb{E} \sup_{\mathbf{C} \in \mathcal{H}^k} |W(\mathbf{C}, \mu) - W(\mathbf{C}, \mu_n)|$. So we obtain the result in this lemma.

$\qquad\square$

## C   Proof of Theorem 2

*Proof.* Note that

$$
\begin{aligned}
&\mathbb{E}\left[ W(\tilde{\mathbf{C}}_{n,m}, \mu) \right] - W^*(\mu) \\
&\leq \underbrace{\mathbb{E}\left[ W(\tilde{\mathbf{C}}_{n,m}, \mu) - W(\tilde{\mathbf{C}}_{n,m}, \mu_n) \right]}_{\text{Term-A}} + \underbrace{\mathbb{E}\left[ W(\tilde{\mathbf{C}}_{n,m}, \mu_n) - W(\mathbf{C}_n, \mu_n) \right]}_{\text{Term-B}} \\
&\quad + \underbrace{\mathbb{E}\left[ W(\mathbf{C}_n, \mu_n) - W(\mathbf{C}_n, \mu) \right]}_{\text{Term-C}} + \underbrace{\mathbb{E}\left[ W(\mathbf{C}_n, \mu) \right] - W^*(\mu)}_{\text{Term-D}}.
\end{aligned}
\tag{58}
$$

According to Lemma 8, with probability $1 - \delta$, we have **Term-A** $\leq \mathcal{O}\left(\frac{\sqrt{k}\log^2(\sqrt{n}) + \sqrt{\log\frac{1}{\delta}}}{\sqrt{n}}\right)$.

According to Lemma 6, for $m = \Omega\left(\frac{4\log n - 2\log \delta}{\varepsilon - \log(1+\varepsilon)}\right)$, we have, with probability at least $1 - \delta$, **Term-B** $\leq \frac{2\varepsilon}{1-\varepsilon}$.

Note that **Term-C** $= \mathbb{E}\left[W(\mathbf{C}_n, \mu_n) - W(\mathbf{C}_n, \mu)\right] \leq \mathbb{E}\sup_{\mathbf{C} \in \mathcal{H}^k} |W(\mathbf{C}, \mu_n) - W(\mathbf{C}, \mu)|$. Therefore, according to Eq.(57), we can obtain **Term-C** $\leq \mathcal{O}\left(\frac{\sqrt{k}\log^2(\sqrt{n}) + \sqrt{\log\frac{1}{\delta}}}{\sqrt{n}}\right)$.

According to Theorem 1, with probability at least $1 - \delta$, we have, **Term-D** $\leq \mathcal{O}\left(\sqrt{\frac{k}{n}}\log^2(\sqrt{n})\right)$.

Here, we complete this proof.

$\square$

## D  Proof of Theorem 3

*Proof.* We have

$$\mathbb{E}_{\mathcal{S}}\left[\mathbb{E}_{\mathcal{J}}\left[W(\mathbf{C}_{n,m}^+, \mu)\right]\right] = \mathbb{E}_{\mathcal{S}}\left[\mathbb{E}_{\mathcal{J}}\left[W(\mathbf{C}_{n,m}^+, \mu)\right] - \mathbb{E}_{\mathcal{J}}\left[W(\mathbf{C}_{n,m}^+, \mu_n)\right]\right] + \mathbb{E}_{\mathcal{S}}\left[\mathbb{E}_{\mathcal{J}}\left[W(\mathbf{C}_{n,m}^+, \mu_n)\right]\right]. \tag{59}$$

According to Lemma 1, one can obtain that

$$\mathbb{E}_{\mathcal{S}}\left[\mathbb{E}_{\mathcal{J}}\left[W(\mathbf{C}_{n,m}^+, \mu_n)\right]\right] \leq \varpi \cdot \mathbb{E}[W(\tilde{\mathbf{C}}_{n,m}, \mu_n)] = \varpi \cdot \mathbb{E}[W(\tilde{\mathbf{C}}_{n,m}, \mu_n) - W(\tilde{\mathbf{C}}_{n,m}, \mu)] + \varpi \cdot \mathbb{E}[W(\tilde{\mathbf{C}}_{n,m}, \mu)],$$

Therefore, Eq.(59) can be transferred into

$$\mathbb{E}_{\mathcal{S}}\left[\mathbb{E}_{\mathcal{J}}\left[W(\mathbf{C}_{n,m}^+, \mu)\right]\right] \leq \underbrace{\mathbb{E}_{\mathcal{S}}\left[\mathbb{E}_{\mathcal{J}}\left[W(\mathbf{C}_{n,m}^+, \mu)\right] - \mathbb{E}_{\mathcal{J}}\left[W(\mathbf{C}_{n,m}^+, \mu_n)\right]\right]}_{\textbf{Term-A}}$$
$$+ \varpi \cdot \underbrace{\mathbb{E}[W(\tilde{\mathbf{C}}_{n,m}, \mu_n) - W(\tilde{\mathbf{C}}_{n,m}, \mu)]}_{\textbf{Term-B}}$$
$$+ \varpi \cdot \underbrace{\mathbb{E}[W(\tilde{\mathbf{C}}_{n,m}, \mu)]}_{\textbf{Term-C}},$$

Note that **Term-A** $\leq \mathbb{E}\sup_{\mathbf{C} \in \mathcal{H}^k} |W(\mathbf{C}, \mu_n) - W(\mathbf{C}, \mu)|$. Therefore, according to Eq.(57), we can obtain **Term-A** $\leq \mathcal{O}\left(\frac{\sqrt{k}\log^2(\sqrt{n}) + \sqrt{\log\frac{1}{\delta}}}{\sqrt{n}}\right)$.

According to Lemma 8, with probability $1 - \delta$, we have **Term-B** $\leq \mathcal{O}\left(\frac{\sqrt{k}\log^2(\sqrt{n}) + \sqrt{\log\frac{1}{\delta}}}{\sqrt{n}}\right)$.

According to Theorem 2, we have **Term-C** $= \mathbb{E}\left[W(\tilde{\mathbf{C}}_{n,m}, \mu)\right] \leq W^*(\mu) + \tilde{\mathcal{O}}\left(\sqrt{\frac{k}{n}}\right) + \mathcal{O}\left(\frac{\varepsilon}{1-\varepsilon}\right)$.

We complete this proof.

$\square$