# OpenReview forum: "Randomized Sketches for Clustering: Fast and Optimal Kernel $k$-Means"
_NeurIPS.cc/2022/Conference — NeurIPS 2022 Accept_

### Official Review · Reviewer_sN6c · 2022-07-08

**Rating:** 9
**Confidence:** 5
**Soundness:** 3 good
**Presentation:** 3 good
**Contribution:** 4 excellent

**Summary:**

The paper investigates the statistical analysis and computational requirements of kernel k-means based on randomized sketches.
The authors proposed a randomized sketches framework for kernel k-means and construct three specific examples of sub-Gaussian sketches, ROS sketches, and Nystrom to kernel k-means.
The proposed methods obtain the optimal excess risk upper bound $\mathcal{O}(\sqrt{k/n})$ with the least sketch dimension $\Omega(\sqrt{n})$ in the theoretical analysis.
Meanwhile, they reduce the time cost to $\mathcal{O}(n\sqrt{n}+n\sqrt{n}kt)$
and the space complexity to $\mathcal{O}(n\sqrt{n})$, which make a good improvement in computational requirements.
Finally, this work adopts the improved kernel k-means++ for the proposed randomized sketches kernel k-mean and derives the similar bound.
The experimental results on both simulated data and real-world data
verified the theoretical results of the proposed methods (sub-Gaussian sketches kernel k-means, ROS sketches kernel k-means, and Nystrom kernel k-means).

**Questions:**

A solid theoretical analysis is provided. A clear comparison with the results of recent works is given in Table 1. The experiment results on synthetic and real data show the effectiveness of the proposed methods. Nevertheless, there are some minor issues.

1) The experiments in this paper are convincing. It would be much better if the figures are more beautiful.

2) The author should double check the text. There are few typos. For example, in Table 1, the citations are connected with the text. "real-word" should be "real-world".

3) In line 200, it is a little bit confusing that the authors present $K$ followed by $\hat{K}$

4) The conclusion section of this paper is missing.

**Limitations:**

This paper describes the scope of application of the proposed methods in detail, and does not involve negative social impact.

**Strengths And Weaknesses:**

The paper has the following contributions:
1) proposed a unified randomized sketches framework to kernel k-means;
2) proposed three novel and specific examples (sub-Gaussian sketches, the randomized orthogonal system (ROS) sketches, and Nystrom kernel k-means);
3) obtained the state-of-the-art risk bounds of the proposed methods in theoretical analysis;
4) made significant improvement in reducing computational requirements,
5) derived the similarity optimal bound of approximate solutions in k-means++.

Those contributions are novel and significant. The experimental evaluations in simulated and real case settings illustrate the theoretical analysis of the proposed methods. The paper is well organized, well written, and technically sound. The exposed materials can be followed up fairly. All the relevant papers are properly cited. Overall I enjoyed reading this paper, which nicely generalizes randomized sketches to the kernel k-means.

---

> ### Author Response · Authors · 2022-07-29
> **Response to Reviewer sN6c**
>
> Thank you for your appreciation and hard review. We strive to improve this paper according to each opinion of the reviewer.
>
> **Q1. The experiments in this paper are convincing. It would be much better if the figures are more beautiful.**
>
> Answer: We update the figures in Figure 1 to show them in a more beautiful form.
>
> **Q2. The author should double check the text. There are few typos. For example, in Table 1, the citations are connected with the text. "real-word" should be "real-world".**
>
> Answer: We carefully examine the full text and correct the typos. We cited Table 1 in Remark 3.
>
> **Q3. In line 200, it is a little bit confusing that the authors present $\mathbf{K}$ followed by $\mathbf{\hat{K}}$.**
>
> Answer: We change the description to avoid misunderstanding in section 3.2.
>
> **Q4. The conclusion section of this paper is missing.**
>
> Answer: We add the conclusion section in the main paper.

---

> > ### Comment · Reviewer_sN6c · 2022-08-07
> > **Thanks for the authors' responses.**
> >
> > Having read the other reviews and responses, I still think this is a very good theoretical paper. The excess risk bounds of kernel clustering have been studied for decades, but none of them have obtained the optimal risk bound with the least approximate dimension and no strong assumptions. This research result has important significance and contribution to some research in the field of clustering. It is possible to promote and guide the future theoretical research in the unsupervised field. Of course, as other reviewers have said, the presentations of this paper needs to be improved, but it does not affect reading on the whole. I remain enthusiastic about this paper and I will keep my original rating.

---

### Official Review · Reviewer_ZBXn · 2022-07-09

**Rating:** 6
**Confidence:** 3
**Soundness:** 3 good
**Presentation:** 3 good
**Contribution:** 3 good

**Summary:**

This paper studies statistical properties and computational efficiency of randomized sketching for kernel k-means clustering. Compared to most existing works, this paper focuses on characterizing the optimal excess clustering risk upper bounds for three types of randomized sketching. In this paper, randomized sketching involves multiplying the kernel matrix by a "random" matrix (also called a sketch matrix). Overall, the paper provides a detailed overview of background materials (i.e., kernel k-means and randomized sketching).

**Questions:**

Can you please explain how Section 4.1 relates to the other sections? It looks like the main contribution of this subsection is to include some results from (Lattanzi & Sohler 2019). However, the significance of this new result has not been discussed in other places, such as experimental results.

**Limitations:**

There should be some more discussion on the significance of the statistical analysis for practitioners. Although this work provides an interesting upper bound, how does this result translate for analyzing large data sets? Can this approach be used for clustering a data set with a few hundreds of thousands of samples?

**Strengths And Weaknesses:**

Strengths:
+ The main strength of this work is related to the provided statistical analysis and upper bounds for the the optimal excess clustering risk. Based on the provided table, this work shows that sketch dimension can be substantially reduced without compromising accuracy.
+ Another strength of this work is providing a nice overview of kernel k-means clustering and randomized sketching. Although these materials are very standard, it is nice to provide them for general readers.
+ This work also presents a "unified" algorithm with time/space complexity analysis, which is beneficial.

Weaknesses:
- From a computational standpoint, the contribution of this work is limited. The three randomized sketching methods have been used for various kernel-based methods, including more sophisticated clustering techniques such as spectral clustering.
- Although the improved upper bound is interesting, experimental results do not directly show the benefits of this statistical analysis. In other words, it is shown that the proposed randomized methods are more efficient compared to the exact kernel k-means, but no comparison with the other cited works.

---

> ### Author Response · Authors · 2022-08-01
> **Response to Reviewer ZBXn (Part-2)**
>
> **Question 3: Can you please explain how Section 4.1 relates to the other sections? It looks like the main contribution of this subsection is to include some results from (Lattanzi Sohler 2019). However, the significance of this new result has not been discussed in other places, such as experimental results.**
>
> Answer: Kernel k-means++ has been proved to be a good application of kernel k-means (Lattanzi Sohler (2019)). Therefore, in section 4.1, we generalize the proposed randomized sketches to  kernel k-means++. We make a theoretical analysis of the randomized sketches kernel k-means++, which is an extension of our theory.
>
> Lemma 1 is a multiplicative error bound on the empirical risk from (Lattanzi Sohler (2019)). Based on Lemma 1, we obtain our Theorem 3 about the proposed randomized sketches kernel k-means++. This results show that, if the optimal clustering risk $W^{\ast}(\mu)$ is smaller than $\tilde{\mathcal{O}}(\sqrt{k/n})$, the proposed randomized sketches kernel k-means++ can also reach the optimal risk bound $\mathcal{\tilde{O}}(\sqrt{k/n})$.
>
> If the reviewer thinks it is irrelevant, we can consider removing this part or putting it in the appendix.
>
> **Limitations 4: There should be some more discussion on the significance of the statistical analysis for practitioners. Although this work provides an interesting upper bound, how does this result translate for analyzing large data sets? Can this approach be used for clustering a data set with a few hundreds of thousands of samples?**
>
> Answer: In the experiments, we add the experiments on the large covtype datasets. If the training time exceeds 90 seconds or the memory is insufficient, the experiment will be stopped. In the large covtype dataset, kernel k-means, Gaussian, and Nytrom (Liu) cannot achieve the experimental results, but our proposed methods can obtain small training time and good accuracy. Those verify  that the proposed methods are more suitable for large-scale samples.
>
> Theoretical analysis show that the proposed methods can obtain the optimal risk bound with the small sketch dimension $m=\Omega(\sqrt{n})$ (in Theorem 2). The time complexity and space complexity are $\mathcal{O}(\sqrt{n}m^2+nmkt)$ and $\mathcal{O}(nm)$ in sub-Gaussian sketches, $\mathcal{O}(nm\log m+nmkt)$ and $\mathcal{O}(nm)$ in ROS sketches, $\mathcal{O}(nmkt)$ and $\mathcal{O}(nm)$ in the proposed Nystr\"{o}m (in Section 3.2), which are related to the sketch dimension $m$. Compared with other methods in this field (in Table 1, we substitute the specific value of $m$ into the expression of time and space complexity), the proposed methods can greatly reduce the time and space complexity with the optimal risk bound. This is one of the innovations of this paper.

---

> > ### Comment · Reviewer_ZBXn · 2022-08-07
> > **Thank you for the response.**
> >
> > Thank you for your detailed response and the revision of the manuscript. My score/rating has been updated. The main contribution of this work is obtaining risk bounds, which I agree that they are interesting theoretical results in the context of clustering. However, as we can see from Table 2, the accuracy results are not necessarily state-of-the-art, which is expected because the practicality of kernel k-means is limited in practice.

---

> ### Author Response · Authors · 2022-08-01
> **Response to Reviewer ZBXn (Part-1)**
>
> Thank you very much for your constructive suggestions.
>
> **Weakness 1: From a computational standpoint, the contribution of this work is limited. The three randomized sketching methods have been used for various kernel-based methods, including more sophisticated clustering techniques such as spectral clustering.**
>
> Answer: Randomized sketches have a long history and have been studied by many papers. The form of  R=SQ is a classic form and not our first. It has been used in many papers. However, there is no works to obtain the optimal excess risk bound in randomized sketches. In this paper, we mainly focus on the theoretical analysis of randomized sketching kernel k-means, deriving the optimal excess risk bound, and filling the defects of unsupervised learning. Meanwhile, we provide a unified randomized sketches framework with greatly reducing the computational requirements.
>
> In theoretical analysis, at present, few of the optimization-based methods focus on the underlying excess risk problem of kernel k-means. To the best of our knowledge, the only three results providing excess risk guarantees for approximate kernel k-means are Biau et al. (2008), Calandriello & Rosasco (2018), and Liu (2021). In Biau et al. (2008), they employ the randomized sketches method to project the data in Hilbert space so as to approximate kernel k-means. In Calandriello & Rosasco (2018), the excess risk upper bound of Nystr\"{o}m is $\mathcal{O}(k/\sqrt{n})$ when the approximate dimension reaches $\Omega(\sqrt{n})$. The excess risk bounds of Biau et al. (2008) and Calandriello & Rosasco (2018) are both linearly dependent on *k* and thus do not obtain the optimal risk bound. In Liu (2021), the approximate Nystr\"{o}m kernel $k$-means obtains the risk upper bound $\mathcal{O}(\sqrt{k/n})$ with the approximate dimension $\Omega(\sqrt{nk})$. Although this paper Liu (2021) further reduces the approximate dimension to $\Omega(\sqrt{n})$ by introducing a stronger assumption, this is not universal. In these three papers, only the method in Biau et al. (2008) is based on randomized sketches, and it is most relevant to our paper. The methods in Calandriello & Rosasco (2018) and Liu (2021) are Nystr\"{o}m. The proposed methods in our paper obtain the optimal excess clustering risk upper bound $\mathcal{O}(\sqrt{k/n})$ with the approximate dimension of $\Omega(\sqrt{n})$. To the best of our knowledge, this is the first optimal excess risk bound with the least approximate dimension and no strong assumptions for general approximate kernel k-means.
>
> In algorithms, we provide a unified randomized sketches framework to kernel k-means. And the specific randomized sketches kernel k-means are different from the previous works. In this field (randomized sketches kernel k-means), only Biau et al. (2008) study excess risk bound which is the most relevant to our paper. Compared to Biau et al. (2008), its sketch matrix is a dense and unstructured Gaussian matrix without sampling matrix. It cannot take advantage of FFT or sparsity to speed up kernel k-means computation. So, its computational requirements is high. The other randomized sketches kernel k-means do not obtain the excess risk bounds, which are different from this paper. Here, we take the classical randomized sketches kernel k-means in Wang et al. (2019) as an example to compare with them. Their algorithm 1 first exacts features of kernel matrix by iteration then reduces the features through orthogonalization. Their algorithm 2 uses a rank-restricted random Nystr\"{o}m approximation. Their Gaussian projection matrix is dense. They are different from our proposed methods. In computational complexity, they provide five methods to construct the sketch matrices. Their smallest time complexity is $\mathcal{O}(nmkt+nc^2+ncd)$ and space complexity is $\mathcal{O}(nc)$, where $c>m$. We propose three specific randomized sketches methods. Our the largest time complexity is $\mathcal{O}(nmkt+\sqrt{n}m^2)$ and space complexity is $\mathcal{O}(nm)$. Therefore, our time and space complexity are always better than theirs under the same $m$.
>
> **Weakness 2: Although the improved upper bound is interesting, experimental results do not directly show the benefits of this statistical analysis. In other words, it is shown that the proposed randomized methods are more efficient compared to the exact kernel k-means, but no comparison with the other cited works.**
>
> Answer: In the experiments of real-world datasets, we add the new experimental results of Nytr\"{o}m  Liu (2021) and Gaussian Biau et al. (2008) in Table 2. From the experimental results on real datasets, we can find that the proposed methods, Nytr\"{o}m Liu (2021), and Gaussian give a similar accuracy as the exact kernel k-means. The proposed methods outperform Nytr\"{o}m Liu (2021) and Gaussian in time cost, which also matches our theoretical analysis.

---

### Official Review · Reviewer_b1Eg · 2022-07-15

**Rating:** 5
**Confidence:** 3
**Soundness:** 3 good
**Presentation:** 2 fair
**Contribution:** 3 good

**Summary:**

This paper proposes a new sketching method for vectors, which allows one to retain the guarantees associated with k-means but with dimension bounded by sqrt{n}, which can be much less than d. The framework can be applied using three standard random matrices (sub-Gaussian, ROS or Nystrom), and of course the reduced dimension allows for better algorithmic runtime.

The core contribution is the creation of a sketch of the form SQK, where S is the full random matrix, Q is a sampling matrix (a random submatrix of the identity matrix), and the multiplication of Q by K yields the approximate kernel matrix between the sampled data points and the point set.

**Questions:**

Please address the above concerns.

**Strengths And Weaknesses:**

Strengths: The overall result is a non-trivial improvement for an important problem.

Weaknesses: I felt the presentation could be greatly improved. There are multiple choices made regarding the construction of the sketch (basically, the SQK format used by the authors), without sufficient explanation, overview or discussion of the import of these choices. The steps of the algorithm should be discussed and explained in depth and they are introduced. A similar issue is that the current technique isn't placed in context - what is its relation to earlier used techniques, and in what way is it different. The overall presentation is poor, and also doesn't allow one to judge the novelty of the paper.

--------------------

In response to the authors' rebuttal, I have raised my overall score.

---

> ### Author Response · Authors · 2022-08-01
> **Response to Reviewer b1Eg (Part-2)**
>
> **Q3. A similar issue is that the current technique isn't placed in context - what is its relation to earlier used techniques, and in what way is it different. The overall presentation is poor, and also doesn't allow one to judge the novelty of the paper.**
>
> Answer: This paper focuses on the excess risk bound and computational requirements for kernel k-means.  Although there are many studies on the approximate kernel k-means, these approximate works pay little attention to the excess risk of clusters with the exception of Biau et al. (2008), Calandriello & Rosasco (2018), and Liu (2021). (Not that, in these three papers, only the method in Biau et al. (2008) is based on the randomized sketches. ) For example, the works in Wang et al. (2019) establish the $1+\varepsilon$ relative-error bound for randomized sketches kernel k-means instead of excess risk bound. Therefore, in this paper, we mainly introduce the most related approximate kernel $k$-means with excess risk guarantees. We add this explanation in Section 1.
>
> Here we provide the detail comparison with some randomized sketches kernel k-means.
>
> In Biau et al. (2008), its excess risk bound is linearly dependent on *k* and thus do not obtain the optimal risk bound. Our proposed methods obtain the optimal risk bound $\mathcal{O}(\sqrt{k/n})$ with the approximate dimension of $\Omega(\sqrt{n})$. In algorithms, they employ the randomized sketches method to project the data in Hilbert space. We project the kernel matrix. Our specific randomized sketches kernel k-means are different from the previous works. And we provide a unified randomized sketches framework to kernel k-means. The sketch matrix in Biau et al. (2008) is a dense and unstructured Gaussian matrix without sampling matrix. It cannot take advantage of FFT or sparsity to speed up kernel k-means computation. So, its computational requirements is high. To the best of our knowledge, our proposed methods obtain the optimal excess risk bound with the least approximate dimension and no strong assumptions for the first time, and they greatly reduce the computational complexity.
>
> Compared to the other randomized sketches kernel k-means. They did not obtain the excess risk bounds. This paper focuses on the excess risk bound and obtains the optimal risk bound. Therefore, we mainly introduce the most related approximate kernel k-means with excess risk guarantees in this paper. Here, we take the classical randomized sketches kernel k-means in Wang et al. (2019) as an example to compare with them. In theoretical analysis, Wang et al. (2019) establishe the $1+\varepsilon$ relative-error bound for randomized sketches kernel k-means instead of excess risk bound. In algorithms, their algorithm 1 first exacts features of kernel matrix by iteration then reduces the features through orthogonalization. Their algorithm 2 uses a rank-restricted random Nystr\"{o}m approximation. Their Gaussian projection matrix is dense. They are different from our proposed methods. In computational complexity, they provide five methods to construct the sketch matrices. Their smallest time complexity is $\mathcal{O}(nmkt+nc^2+ncd)$ and space complexity is $\mathcal{O}(nc)$, where $c>m$. We propose three specific randomized sketches methods. Our the largest time complexity is $\mathcal{O}(nmkt+\sqrt{n}m^2)$ and space complexity is $\mathcal{O}(nm)$. Therefore, our time and space complexity are always better than theirs under the same $m$.
>
> To the best of our knowledge, this is the first time that the randomized sketches kernel k-means obtains the optimal excess risk bound with only a fraction of computations.

---

> > ### Author Response · Authors · 2022-08-08
> > **Ask Reviewer b1Eg**
> >
> > Dear reviewer b1Eg, the discussion period is coming to an end. Do my answers address your concerns? If you don't think so, please point out and let me know. I'd be happy to answer any of your concerns.

---

> > ### Comment · Area_Chair_sdzQ · 2022-08-09
> > **Author rebuttal phase closing today**
> >
> > The author-response phase closes today. Please acknowledge the author rebuttal and state if your position has changed. Thanks!

---

> ### Author Response · Authors · 2022-08-01
> **Response to Reviewer b1Eg (Part-1)**
>
> We are very grateful to reviewer for your comments.
>
> **Q1. I felt the presentation could be greatly improved. There are multiple choices made regarding the construction of the sketch (basically, the SQK format used by the authors), without sufficient explanation, overview or discussion of the import of these choices.**
>
> Answer: In this paper, the proposed sketch matrices are $\mathbf{R}=\mathbf{SQ}$, where $\mathbf{Q}$ is a sampling matrix and $\mathbf{S}$ is a structured (in ROS) or sparse (in sub-Gaussian and Nystr\"{o}m) matrix.
>
> In algorithm, the function of $\mathbf{Q}$ is to reduce the scale of data. The function of $\mathbf{S}$ is to fuse data features.
>
> In complexity, the proposed randomized sketches can reduce the time and space complexity. We sample data points according to the sampling matrix, then generate the variant kernel matrix, instead of generating and processing the kernel matrix directly, which can greatly reduce the time and space complexity. In addition, our matrices $\mathbf{S}$ are structured (in ROS) or sparse (in sub-Gaussian and Nystr\"{o}m), which can speed up kernel k-means by FFT or sparsity. For example, due to the constructed property of the Hadamard matrix  $\mathbf{A}$ in ROS, we can use FFT (Fast Fourier Transform algorithm) to compute the matrix-vector product, such as $\mathbf{Au}$ for any $\mathbf{u}\in \mathbb{R}^{m}$, whose time complexity is $\mathcal{O}(m\log m)$ instead of $\mathcal{O}(m^2)$; We only need to compute the non-zero elements instead of the total elements in the sparse matrix multiplication.
>
> In theoretical analysis, we obtain the optimal excess risk bound with a small sketch dimension $\sqrt{n}$ based on the proposed randomized sketches, which can further reduce the time and space complexity.
>
> Overall, the proposed randomized sketches can greatly reduce the time and space complexity with the optimal excess risk bound.
>
> The form of $\mathbf{R}=\mathbf{SQ}$ is a classic form and not our first. It has been used in many papers. Therefore, we did not explain it in detail in the first edition of the paper. In Section 3.2 of the new edition of this paper, we add the explanations.
>
> **Q2. The steps of the algorithm should be discussed and explained in depth and they are introduced.**
>
> Answer: The proposed algorithm in Algorithm 1 is mainly divided into two parts. The first part is from step 1 to step 4, which is mainly to construct the sketch matrix $\mathbf{R}=\mathbf{SQ}$ and obtain the variant kernel matrix $\tilde{\mathbf{K}}=\mathbf{SQK}$. The second part is from step 5 to step 6, mainly performing k-means over the columns of $\tilde{\mathbf{K}}$ and obtaining centroids. We add the above in Section 3.1.
>
> In step 1, one samples $m$ data points from $\mathcal{S}$ according to the sampling matrix $\mathbf{Q}$. Then, computing the variant kernel matrix $\hat{\mathbf{K}}\in \mathbb{R}^{m\times n}$ by $m$ sampling data points and all $n$ data points.
> From a mathematical point of view, this step can be expressed as $\hat{\mathbf{K}} = \mathbf{QK}$. In step 3, we construct a matrix $\mathbf{S}$, whose specific expression will be given in Section 3.2. This paper provides three different examples of $\mathbf{S}$, which brings different effects in the approximate kernel $k$-mean algorithms. In step 5, take the columns of $\tilde{\mathbf{K}}=\mathbf{S}\hat{\mathbf{K}}$ generated in step 4 as the processing objects and execute $k$-means algorithm on them. Finally, compute the centroids $\tilde{\mathbf{C}}_{n,m}$ in Eq.(11).
>
> The explanations of each step were shown in Section  3.1.

---

### Author Response · Authors · 2022-08-01
**Paper Revision**

We have carefully revised the manuscript based on the initial reviews. The following  is a summary of major changes:

1. In Section 1, we provide an explanation and comparison between the existing randomized sketching kernel k-means and our methods (In response to Question 3 by **Reviewer b1Eg**).
2. In Section 3.1, we provide an in-depth explanation of the algorithm steps (In response to Question 2 by **Reviewer b1Eg**).
3. In Section 3.2, we provide the explanation of the construction of the sketch matrix $\mathbf{R}=\mathbf{SQ}$ (In response to Question 1 by **Reviewer b1Eg**).
4. In Section 5.2, we add the experiments on the related methods of Gaussian and Nystrom (Liu) (In response to Weakness 2 by **Reviewer ZBXn**), add the experiments on the large datasets covtype (In response to Limitations 4 by **Reviewer ZBXn**), and revise Table 2.
5. In Figure 1, we update the figures to show them in a more beautiful form (In response to Q1 by **Reviewer sN6c**).
6. We carefully examine the full text and correct the typos and misunderstanding (In response to Q2 and Q3 by **Reviewer sN6c**).
7. We add the Section of Conclusions (In response to Q4 by **Reviewer sN6c**).

We sincerely appreciate all reviewers for their invaluable and constructive comments. Hopefully, reviewers will be satisfied with these responses.

---

### Author Response · Authors · 2022-08-07
**Response to reviewers**

Dear reviewers,

Please let us know if we addressed your concerns. If not, could you please let us know which concerns were not addressed so that we have a chance to respond before the end of the Author-Reviewer discussion period on August 9? Thank you!

---

### Meta-Review · Area_Chair_sdzQ · 2022-08-29

**Recommendation:** Accept
**Confidence:** Certain

**Metareview:**

The paper analyses the performance of a class of sampling-based sketches for the kernel k-means clustering problem. The main contribution is a proof that the excess clustering risk of these sketches is optimal when a smaller sketch is used than in previous state-of-the-art approaches to sketching kernel k-means. Clear comparisons with the prior state-of-the-art results shows that the method greatly reduces the computational complexity of sketched kernel k-means, and experimental validation shows that the method is competitive in terms of accuracy.

**Award:**

No

---

### Decision · Program_Chairs · 2022-09-14

Accept